# Uncovering hidden brain state dynamics that regulate performance and decision-making during cognition

Jalil Taghia[1], Weidong Cai[1], Srikanth Ryali [1], John Kochalka[1], Jonathan Nicholas [1] Tianwen Chen[1] & Vinod Menon[1,2,3]

Human cognition is influenced not only by external task demands but also latent mental processes and brain states that change over time. Here, we use novel Bayesian switching dynamical systems algorithm to identify hidden brain states and determine that these states are only weakly aligned with external task conditions. We compute state transition probabilities and demonstrate how dynamic transitions between hidden states allow flexible reconfiguration of functional brain circuits. Crucially, we identify latent transient brain states and dynamic functional circuits that are optimal for cognition and show that failure to engage these states in a timely manner is associated with poorer task performance and weaker decision-making dynamics. We replicate findings in a large sample ($N = 122$) and reveal a robust link between cognition and flexible latent brain state dynamics. Our study demonstrates the power of switching dynamical systems models for investigating hidden dynamic brain states and functional interactions underlying human cognition.

[1] Department of Psychiatry & Behavioral Sciences, Stanford University School of Medicine, Stanford, CA 94305, USA. [2] Department of Neurology & Neurological Sciences, Stanford University School of Medicine, Stanford, CA 94305, USA. [3] Stanford Neuroscience Institute, Stanford University School of Medicine, Stanford, CA 94305, USA. These authors contributed equally: Jalil Taghia, Weidong Cai. Correspondence and requests for materials should be addressed to J.T. (email: taghia@stanford.edu) or to W.C. (email: wdcai@stanford.edu) or to V.M. (email: menon@stanford.edu)

Flexible and adaptive human cognition depends on dynamic brain circuits that transiently link distributed brain regions in response to moment-by-moment changes in task demands[1–6]. However, a central unaddressed challenge here is that human cognition is influenced not only by (known) external task demands but also by (unknown) latent mental processes that change with time. Uncovering hidden brain states and their dynamic spatio-temporal evolution in relation to cognitive task demands remains an important unresolved problem in human cognitive neuroscience research[7]. Progress in tackling this challenge has been limited due to a lack of appropriate computational tools for quantitative characterization of hidden brain states and their dynamic functional properties such as state transition probabilities and time-varying functional connectivity. To address this challenge here we develop and apply novel unsupervised learning procedures based on Bayesian switching linear dynamical systems (BSDS) to identify latent brain states and characterize their dynamic spatiotemporal properties. Our approach overcomes major limitations of existing methods for studying dynamic interactions in the human brain and provides a novel integrated framework for identifying latent brain states and dynamic brain connectivity during cognition. The scientific aspect of our investigation focuses on dynamic brain states and circuits associated with frontoparietal cortical regions involved in working memory (WM), a process fundamental to human cognition[8–10].

The investigation of time-varying, context-dependent, brain states is a challenging computational problem because of the inherent complexities of nonlinear and latent dynamical processes that characterize brain function[1,3–6,11]. Importantly, changes in brain states and connectivity can be induced by external stimuli and by latent factors, such as motivation, alertness, fatigue and momentary lapse in attention, which can dramatically impact behavior[12–14]. To address this challenge, we leverage advances in machine learning[15,16] and switching linear dynamical systems models[15,17,18] to identify hidden brain states and dynamic brain connectivity using noninvasive fMRI recordings. We demonstrate that probabilistic models can uncover behaviorally significant hidden functional circuit dynamics in the human brain.

A notable feature of the current study is that our computational approach overcomes several major limitations of existing methods for probing dynamic processes in the human brain[19]. The application of extant approaches to cognitive task-based fMRI is particularly problematic because they do not capture the effects of latent brain processes arising from intrinsic functional connectivity and internal mental processes. Furthermore, previous approaches for characterizing dynamic interactions in the human brain have primarily been based on sliding window and clustering techniques applied to observed resting-state fMRI data[1,19–26]. Consequently, little is known about hidden brain states, their temporal evolution, their underlying dynamic functional circuits, and their relation to behavioral performance during cognitive tasks such as WM. Most previous studies in the field have relied on ad hoc procedures for determining critical parameters, such as the window length and number of brain states, which are known to greatly influence the estimation of dynamic brain states and connectivity[19]. Thus, new models based on unsupervised learning procedures for identifying latent brain states, their temporal evolution, lifetimes and occurrence of states, and their switching probabilities are needed for analysis of latent brain dynamics in human fMRI data. BSDS achieves this by using continuous state-space representations in a nonlinear manifold that are modeled via a system of switching state-space models.

BSDS implements an unsupervised learning algorithm that determines hidden (latent) brain states and dynamic switching processes from observed data. Briefly, each brain state is associated with a unique dynamical process that captures time-varying functional connectivity in an optimal latent subspace. Importantly, BSDS does not require arbitrary moving windows nor does it impose temporal boundaries associated with predefined task conditions. BSDS applies a hidden Markov model (HMM) to latent space variables of the observed data, resulting in a parsimonious model of generators underlying the observed data—this is contrast to previous approaches that have applied HMMs directly to observed MEG[27] and resting-state fMRI[28,29] data. These and other features (Methods) allow BSDS to uncover latent brain states, their temporal evolution, volatility, and persistence over time, probability of transition to other brain states, and non-optimal brain state transitions that impair performance. Finally, the temporal evolution of brain states and the covariance structures of each state can be used to extract moment-by-moment connectivity patterns and dynamic functional networks associated with each brain state.

We applied BSDS and other novel computational analyses to fMRI data obtained during a WM task that required participants to switch between different levels of cognitive load. WM, the ability to maintain and manipulate information in the absence of sensory input, is a fundamental component of a wide range of cognitive tasks[8,9]. In virtually all analyses of such cognitive tasks with time-varying cognitive load it is assumed that brain states are perfectly aligned with experimentally determined task parameters. The lack of appropriate computational tools has precluded examination of hidden brain states underlying WM, the transition between task conditions, and individual differences in engagement of latent brain states.

We leveraged high-temporal resolution fMRI data (sampling time = 0.72 s) acquired during an $n$-back WM task from multiple sessions across a large cohort of adults ($N = 122$) who participated in the Human Connectome Project (HCP, http://www.human connectome.org/). We focused our analyses on key nodes of the salience, central executive, and default mode networks (SN, CEN, and DMN, respectively), three large-scale cingulo-opercular and frontoparietal neurocognitive networks whose dynamic interactions play an essential role in cognition and WM in particular[9,10,30,31]. Using BSDS we uncovered multiple critical spatiotemporal properties of latent brain states, including (i) occupancy rates and mean lifetimes of task dominant and non-dominant brain states, (ii) transition states and their relation to flexible task switching, (iii) probability of transitions across states, (iv) dynamic functional networks associated with distinct brain states, and (v) brain states that best predict cognitive performance and decision-making dynamics. We identify optimal brain states associated with WM and test the hypothesis that failure to engage them, and switch between different latent states, significantly impairs cognitive performance and context-specific decision-making. Finally, we demonstrate the replicability and robustness of our findings across sessions.

## Results

**Validation of BSDS using simulations and opto-fMRI stimulation.** Figure 1 describes the key elements of our BSDS model; a detailed description of the mathematical formulation and its algorithmic implementation for estimating latent brain states and dynamic brain connectivity is described in Methods (see also Supplementary Methods, Supplementary Fig. 1, Supplementary Tables 1–3).

To validate the BSDS model, the robustness and accuracy of BSDS in identifying brain states and dynamic functional connectivity were first tested in simulated data. We used neuronal mass and spiking network models and examined whether BSDS can accurately identify ON and OFF states in simulated data. Simulations using the neuronal mass model were carried out using The Virtual Brain (TVB)[32], a state-of-the-art platform for

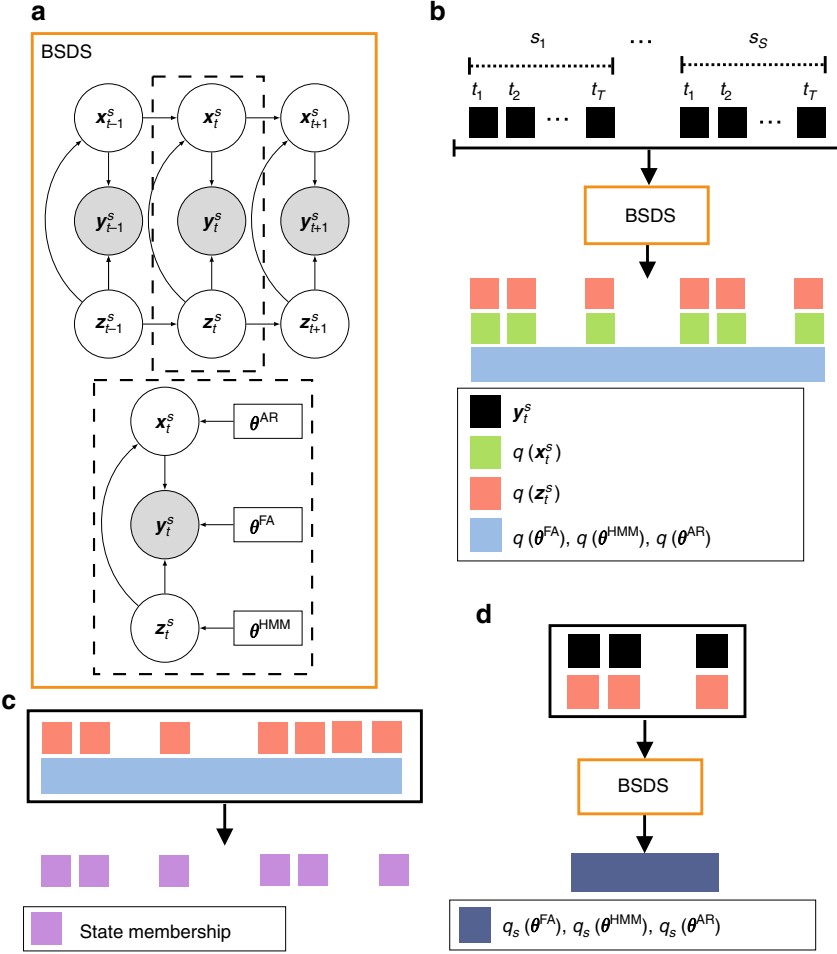

**Fig. 1** The Bayesian switching dynamical systems (BSDS) model and dynamical measures. **a** Graphical representation of the BSDS model (see also Supplementary Fig. 1). $y_t^s$ is the observed vector of ROI time series at time $t$ for subject $s$, $z_t^s$ denotes the latent states that are interdependent through a first-order Markov chain constrained by a hidden Markov model (HMM) parameterized by $\theta^{\mathrm{HMM}}$. In any given state, observations $y_t^s$ are assumed to be generated from a factor analysis (FA) model where $\theta^{\mathrm{FA}}$ indicates FA model parameters and $x_t^s$ are the latent space variables (factor source variables) of the model. Latent variables are further assumed to be generated using an autoregressive (AR) model, where $\theta^{\mathrm{AR}}$ indicates the AR model parameters (Methods). **b** Group-level application of BSDS in which all model parameters, including number of states, are learned primarily from the data. BSDS takes sequences of ROI time series from all subjects as input data. The output of the model is the posterior distribution of all group model parameters, $\{q(\theta^{\mathrm{HMM}}), q(\theta^{\mathrm{FA}}), q(\theta^{\mathrm{AR}})\}$, and posterior distribution of latent state and latent space variables, $q(z_t^s)$ and $q(x_t^s)$, $\forall s$., respectively. **c** Temporal evolution of states indicates state membership at a given time and for a given subject. Temporal evolution of states are computed using posterior distribution of the latent state variables, $q(z_t^s)$, and group model parameters (specifically: $q(\theta^{\mathrm{HMM}})$) input to a Viterbi decoder. Optimal number of states, their occupancy rates, mean lifetimes, and switching probabilities are extracted from temporal evolution of states (Supplementary Methods). **d** Learning subject-level posterior distribution of the model parameters: $\{q_s(\theta^{\mathrm{HMM}}), q_s(\theta^{\mathrm{FA}}), q_s(\theta^{\mathrm{AR}})\}$ $\forall s = 1, \ldots, S$. To learn the subject-level model parameters, BSDS was applied to data from a given subject in an informative fashion. BSDS was initialized using posterior distribution of the latent sate and latent space variables that were previously learned from **b**. BSDS then combined priors with the subject data to learn the posterior distribution of its model parameters. Moment-by-moment covariance structures, $\forall t, s$, were then obtained from temporal evolution of states and subject-level covariance matrices for each state (Supplementary Methods)

large-scale brain network modeling. Simulations utilized a dynamic mean field model[33] which approximates the temporal dynamics of a population of spiking neurons. Recent work has demonstrated the efficacy of such models in emulating important features of human brain activity[34,35]. We performed several simulations of a three-node network using different random initial conditions and noise streams. Each simulation run contained five 60 s ON/OFF task cycles. Task was modeled by stimulating node 1, providing 0.1 nA current with 5 ms pulse width at a rate of 50 Hz for the duration of each 20 s ON block (Fig. 2a). We applied BSDS to the resulting time series and found that it could precisely estimate the onset and offset of each simulated ON/OFF task block based on their unique connectivity patterns.

Next, we examined whether BSDS could uncover latent states in a more realistic neurobiological circuit with a complex mix of excitatory–inhibitory interactions between neuronal ensembles. Here we leveraged prior work on a standard cortical-basal ganglia-thalamus circuit utilizing Nengo[36]. Seven nodes of this circuit encompassing the basal ganglia (globus pallidus internal, globus pallidus external, subthalamic nucleus, striatum D1, striatum D2), thalamus and motor cortex were constructed using ensembles of leaky integrate-and-fire neurons[37]. To mimic multiple task conditions with individual (subject) variation, three different 16-dimensional input vectors were selected for each (subject). As with the TVB model, Nengo-based simulations accurately uncovered the onset and offset of each simulated block based on their distinct connectivity patterns (Fig. 2b). These

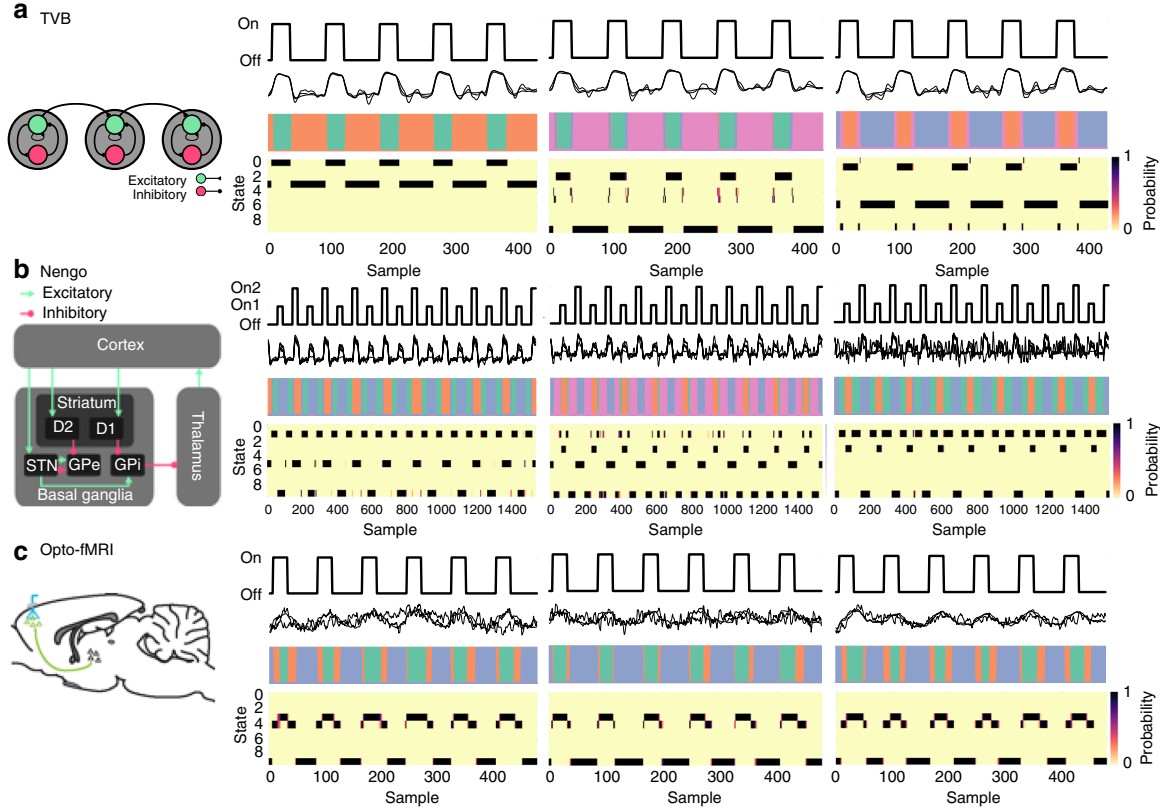

**Fig. 2** Illustration of dynamic brain states that were automatically estimated using BSDS. In both simulated and experimental opto-fMRI data, BSDS accurately and reliably identified distinct brain states across time. **a** Data simulated using neurophysiological realistic models implemented in The Virtual Brain (TVB); **b** Data simulated using a cortico-thalamic-basal ganglia model implemented in Neural Engineering Object (Nengo). In each simulation, three exemplar data (subjects) and results are presented in the three sub-panels from left to right. In each sub-panel, the top row illustrates task waveform (e.g., stimulation on and off states); the second row shows simulated time series; the third row shows the temporal evolution of each latent state; and the bottom row shows posterior probabilities of each states across time. Note that for **a** and **b** BSDS was applied in a subject-wise fashion. **c** Dynamic brain states determined from experimental opto-fMRI data acquired during in vivo optogenetic stimulation in three rodents. In addition to ON and OFF states, BSDS also identified a novel transition state associated with switching between the ON and OFF states. Here, BSDS was applied in a group-wise fashion. The rodent brain diagram in the left column is adapted from a previous study[39]

results demonstrate the robustness and accuracy of BSDS in identification of brain states and functional connectivity in neurobiologically realistic simulated data.

To further illustrate the power of BSDS in uncovering latent brain dynamics we used whole-brain rodent optogenetic fMRI (opto-fMRI) data acquired during stimulation of primary motor cortex (M1)[38,39] (see Methods for details of data acquisition and analysis). Opto-fMRI is an optimal tool here as it allows for probing the direct effects of in vivo brain stimulation to characterize dynamic brain states and functional connectivity between stimulated and downstream target brain regions. fMRI time series were extracted from M1, the thalamus, a downstream target, and the insula, a non-activated control region[39]. When applied to time series extracted from these regions BSDS revealed three distinct states: as expected, we detected robust ON and OFF states that were consistently aligned, but not completely defined, by the external optogenetic stimulation protocol (Fig. 2c). More interestingly, BSDS revealed a third state that appeared immediately before and after the onset of the ON state (Fig. 2c). Thus, BSDS not only uncovers task-induced brain states, but also latent transition states and their temporal boundaries.

**Latent brain states during WM and task-boundary alignment**

**WM.** To address the main scientific goal of our study, we characterized latent brain dynamics, including latent states, their

occupancy rates and transition probabilities, relation to task onset and offset, dynamic functional connectivity and relation to task boundaries (Fig. 3c) using high-temporal resolution fMRI data from the HCP. These data were acquired in two separate sessions and each session included a run with multiple blocks of a 2-back WM task, a 0-back control task and a passive fixation (rest) condition (Fig. 4b)[40]. Data from the two sessions were analyzed separately to determine robustness and replicability of our findings.

We applied BSDS to cingulo-opercular, lateral frontoparietal, and default mode network regions that showed the strongest WM load-dependent activations and deactivations in this group of 122 participants. Consistent with previous studies of WM, this included key nodes of the SN, CEN, and DMN: bilateral anterior insula (AI), middle frontal gyrus (MFG), frontal eye field (FEF), intraparietal sulcus (IPS), right dorsomedial prefrontal cortex (DMPFC), left ventromedial prefrontal cortex (VMPFC), and posterior cingulate cortex (PCC) (Fig. 3a, b).

Next, we used BSDS and classification analysis to probe latent brain dynamics associated with these network nodes, including distinct brain states, their fluctuations over time and their relation to experimental manipulation of WM load. BSDS isolated four dynamic brain states, each with a distinct pattern of covariance across SN, CEN, and DMN nodes, and brain states were matched between the two sessions (Methods; Supplementary Tables 4, 5). The four brain states were only partially aligned with onset and

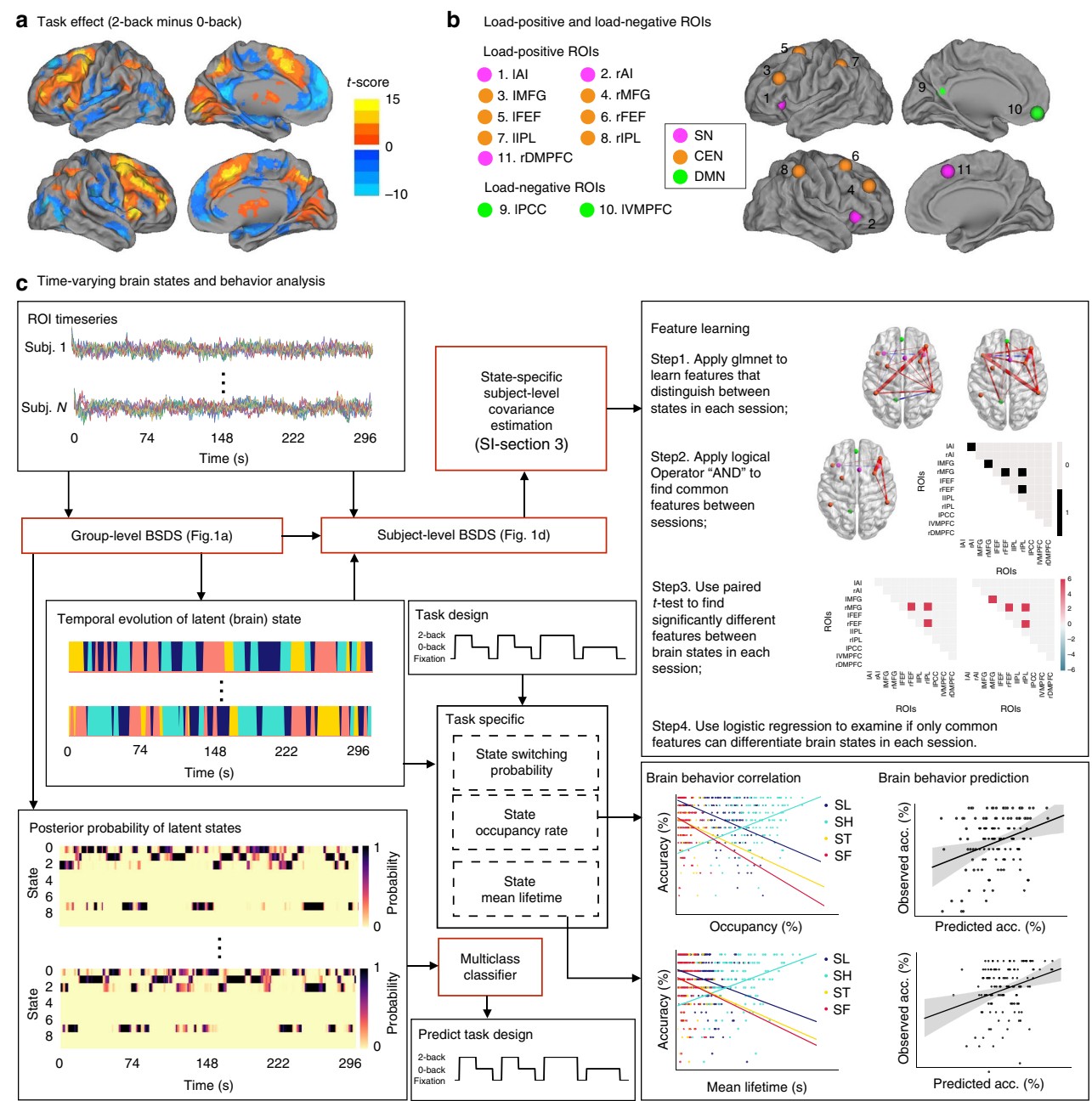

**Fig. 3** Application of BSDS to human WM data from the Human Connectome Project. **a** Brain regions activated (warm colors) and deactivated (cool colors) during the 2-back, compared to the 0-back, task condition. **b** Regions of interest (ROIs) were determined using activation and deactivation peaks from **a**. ROIs activated during WM predominantly overlapped with the Salience Network (SN) and Central Executive Network (CEN): 1, left anterior insula (lAI); 2, right anterior insula (rAI); 3, left middle frontal gyrus (lMFG); 4, right middle frontal gyrus (rMFG); 5, left frontal eye field (lFEF); 6, right frontal eye field (rFEF); 7, left intraparietal lobule (lIPL); 8, right intraparietal lobule (rIPL); and 11, right dosomedial prefrontal cortex (rDMPFC). ROIs deactivated during WM mainly overlapped with the Default Mode Network (DMN): 9, left posterior cingulate cortex (lPCC) and 10, left ventromedial prefrontal cortex (lVMPFC). **c** Schematic illustration of time series analysis. We first applied group-level BSDS on the ROI time series. We then performed the following post-analyses. In the first part of the analysis, we used the group-level estimated temporal evolution of states together with the group-level learnt model parameters as the input for subject-level BSDS analysis for estimation of subject-specific and state-specific covariance matrices. A feature learning procedure was then used to discriminate latent states based on dynamic network features, computed from covariance matrices. In the second part of the analysis, we first extracted dynamic state properties: occupancy rate, mean lifetime, and transition probabilities of states from the estimated temporal evolution of states. We then performed brain behavior analysis using these dynamic state properties. In the third part of the analysis, we used a multiclass classifier trained on the estimated posterior probability of latent states to predict task waveform

offset of the three experimental task conditions (Fig. 4a, b). Analysis of the moment-by-moment correspondence between the posterior probabilities of individual brain states and the actual 2-back, 0-back, and fixation (rest) conditions using a multiclass classifier revealed within-session and cross-session cross-validation accuracies that significantly exceeded the chance level of 33%. However, prediction accuracies ranged from 49 to 55%, far less than would be expected from a complete alignment with temporal boundaries of task onset and offset (Supplementary Fig. 2).

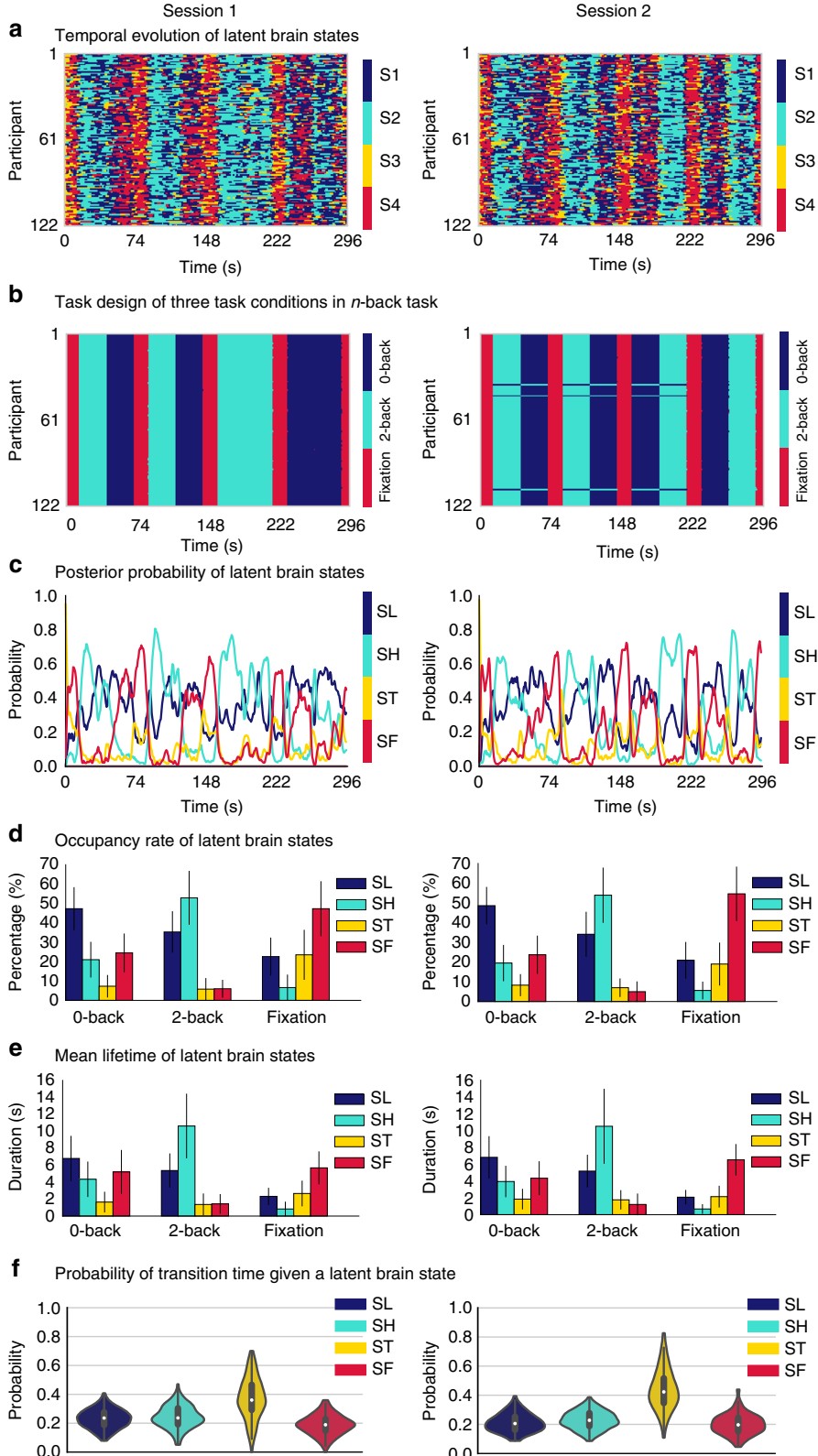

**Fig. 4** Latent brain states during WM, their dynamic properties. **a** Temporal evolution of the four latent brain states identified in each of the 122 participants. **b** Corresponding task waveforms of the three task conditions in the *n*-back WM task—0-back, 2-back, and fixation blocks—are shown in the same layout. **c** Time-varying posterior probability of each latent brain state across participants. **d** Occupancy rates of latent brain states for the three states which dominate the 0-back, 2-back, and fixation task blocks, SL, SH, and SF respectively (*p*s < 0.001, two-tailed *t*-test). **e** Mean lifetimes of latent brain states for the three states which dominate the 0-back, 2-back, and fixation task blocks, SL, SH, and SF respectively (*p*s < 0.001, two-tailed *t*-test). **f** BSDS revealed a novel transition state ST, which occurs more frequently right after the onset of experimental task blocks (*p*s < 0.001, two-tailed *t*-test). Note that SL, SH, ST, and SF were named by their task dominancy in **c–f**, which correspond to S1, S2, S3, S4 in **a**. Error bar stands for standard deviation. Color code mapping in **a**, **c**, **d**, **e**, and **f**: dark blue—SL, cyan—SH, yellow—ST, red—SF. Color code mapping in **b**: dark blue—0-back, cyan—2-back, red—fixation

**Occupancy of latent brain states during WM**. The 2-back, 0-back, and fixation conditions were each dominated by distinct brain states designated SH (high-load state), SL (low-load state), and SF (fixation state) respectively (Fig. 4d). The brain state dominant during the 2-back WM task condition had an occupancy rate of $52.9 \pm 13.9\%$ in session 1, while the other three non-dominant states had occurrence rates of $35.3 \pm 10.7$, $5.8 \pm 5.7$, and $6.0 \pm 4.6\%$, respectively. In both Sessions, the occurrence of the dominant state during the 2-back condition was significantly higher than the occurrence of other non-dominant states (all $ps < 0.001$, two-tailed $t$-test). The mean lifetime of the dominant state in the 2-back condition was $10 \pm 4$ s in both sessions, significantly longer than the mean lifetime of the other non-dominant brain states (all $ps < 0.001$, two-tailed $t$-test), but much shorter than the 27.5 s task block (Fig. 4e). Thus, the 2-back condition is characterized by a mixture of brain states, with the dominant state active for only a relatively short interval. A similar pattern was observed for the states that were dominant in the 0-back and fixation conditions (Fig. 4d, e). These results demonstrate that the WM task is characterized by latent task-induced states whose fractional occupancy is relatively short compared to the task blocks.

**A novel transition state during WM**. In addition to SH, SL, and SF, BSDS uncovered a fourth (transition) state (ST in Figs. 4 and 5). The occupancy rate and mean lifetime of this state during the 2-back, 0-back, and fixation conditions was lower than the three other states ($<25\%$ and 3 s, respectively). ST was more likely to occur during transition after the onset of new task blocks ($32 \pm 16\%$ in session 1 and $44 \pm 14\%$ in session 2, Fig. 4f, Supplementary Fig. 3), significantly higher than other latent states in both sessions (all $ps < 0.001$, two-tailed $t$-test). These results demonstrate that cognitive tasks with multiple conditions are characterized not only by latent task-induced states but also by transition states.

**Transition probabilities of latent brain states during WM**. Next, we used BSDS to investigate dynamic temporal properties of transitions between hidden brain states. A powerful feature of BSDS is that it provides moment-by-moment estimates of the probability of either switching between latent states or staying within the same brain state. We computed the state transition matrix for each participant and first examined the likelihood that a brain state at time instance $t$ remained within the same brain state in the previous time $t − 1$. This analysis revealed that all four brain states are sticky: i.e., states are not volatile from one-time step to another, they persist over time (Fig. 5), but, as shown above, they also do not persist over the entire durations of task blocks. These findings are significant because they suggest that latent brain states are stable over time.

**State transitions during WM are constrained by switch paths**. We next analyzed dynamic properties associated with switching from one brain state to another. Interestingly, this analysis revealed that while most brain states switch between each other in an equi-probable manner, the dominant states in the high-load 2-back WM task and fixation conditions (SH and SF, respectively) do not directly switch between each other (Fig. 5d). By further examining specific state switching paths (Fig. 5e, Supplementary Table 6), we determined that transitions between SH and SF requires passing through SL or through ST. Thus, the brain state that dominates the high-load WM condition does not suddenly shift to the state that dominates the fixation condition without first accessing a state associated with an intermediate cognitive demand.

**Dynamic latent state brain connectivity changes during WM**. We next examined dynamic functional connectivity patterns associated with each latent brain state (Fig. 6). Multivariate analysis was first conducted to determine patterns of functional connections that differentiated the latent brain states. We found a consistent pattern of connections that differentiated brain states in both Sessions 1 and 2. Each state determined by BSDS was associated with a distinct pattern of functional connectivity between SN, CEN, and DMN nodes (Fig. 6a–d). Importantly, classifiers based on connectivity features accurately and reliably differentiated between all latent brain states (all $ps = 0.002$, permutation test, Supplementary Table 7). Analysis of specific links using univariate link-by-link analysis revealed that SH, which dominates the 2-back condition, compared to SL, which dominates the 0-back task condition, showed stronger connectivity of right front-parietal CEN nodes ($ps < 0.05$, two-tailed $t$-test, FDR-corrected for multiple comparisons, Fig. 6e, f).

**Dynamic brain connectivity of transition states during WM**. Crucially, the transition state, ST, had the strongest connectivity within and between the SN, CEN, and DMN nodes in both sessions ($ps < 0.05$, two-tailed $t$-test, FDR-corrected for multiple comparisons, Fig. 6e, f). Thus, as expected, the state that dominates the 2-back high-load WM condition has higher fronto-parietal connectivity compared to the state that dominates the 0-back low-load condition. The surprising finding here is that it was the novel transition state, ST, identified by BSDS that was highest, with higher and more extensive enhancements in connectivity than even the state SH that dominates the high-load WM condition.

To further characterize the unique functional organization of ST, we conducted network analyses and computed degree and betweenness centrality of each region of interest (ROI) for each latent brain state. We found a significant interaction between region (11 ROIs) and state (four latent states) for both node degree and betweenness centrality ($p < 0.001$, two-tailed $t$-test). Post-hoc $t$-tests revealed that DMPFC has significantly higher node degree than all the other ROIs in state ST (all $ps < 0.001$, two-tailed $t$-test) and significantly higher node betweenness centrality than all the other regions but left FEF (all $ps < 0.01$, two-tailed $t$-test). Furthermore, DMPFC has significantly higher node degree in ST than all the other states (all $ps < 0.001$, two-tailed $t$-test) and has significantly higher betweenness centrality in ST than SL and SF (all $ps < 0.05$, two-tailed $t$-test, Supplementary Fig. 4). Multiple comparisons were corrected using Bonferroni procedure. All of these results were replicated across Sessions 1 and 2.

**Latent brain states predict WM performance**. To probe the relation between latent brain states and task performance, we took advantage of a key feature of BSDS, which provides estimates of moment-by-moment changes in brain states and connectivity. We examined whether time-varying brain state changes could predict WM performance. Specifically, we trained a multiple linear regression model to fit estimated the 2-back accuracy using occupancy rates of brain states in the 2-back condition, applied the model on unseen data to predict accuracy, and evaluated model performance by comparing estimated accuracy and observed value across all the subjects. This analysis revealed a significant relation between predicted and actual accuracy (all $ps < 0.001$, Pearson's correlation, Fig. 7a). Notably, each of these results was replicated in both Sessions 1 and 2, highlighting the robustness of our brain behavior findings.

We then tested the hypothesis that the occupancy rate of individual brain states in the 2-back condition is associated with

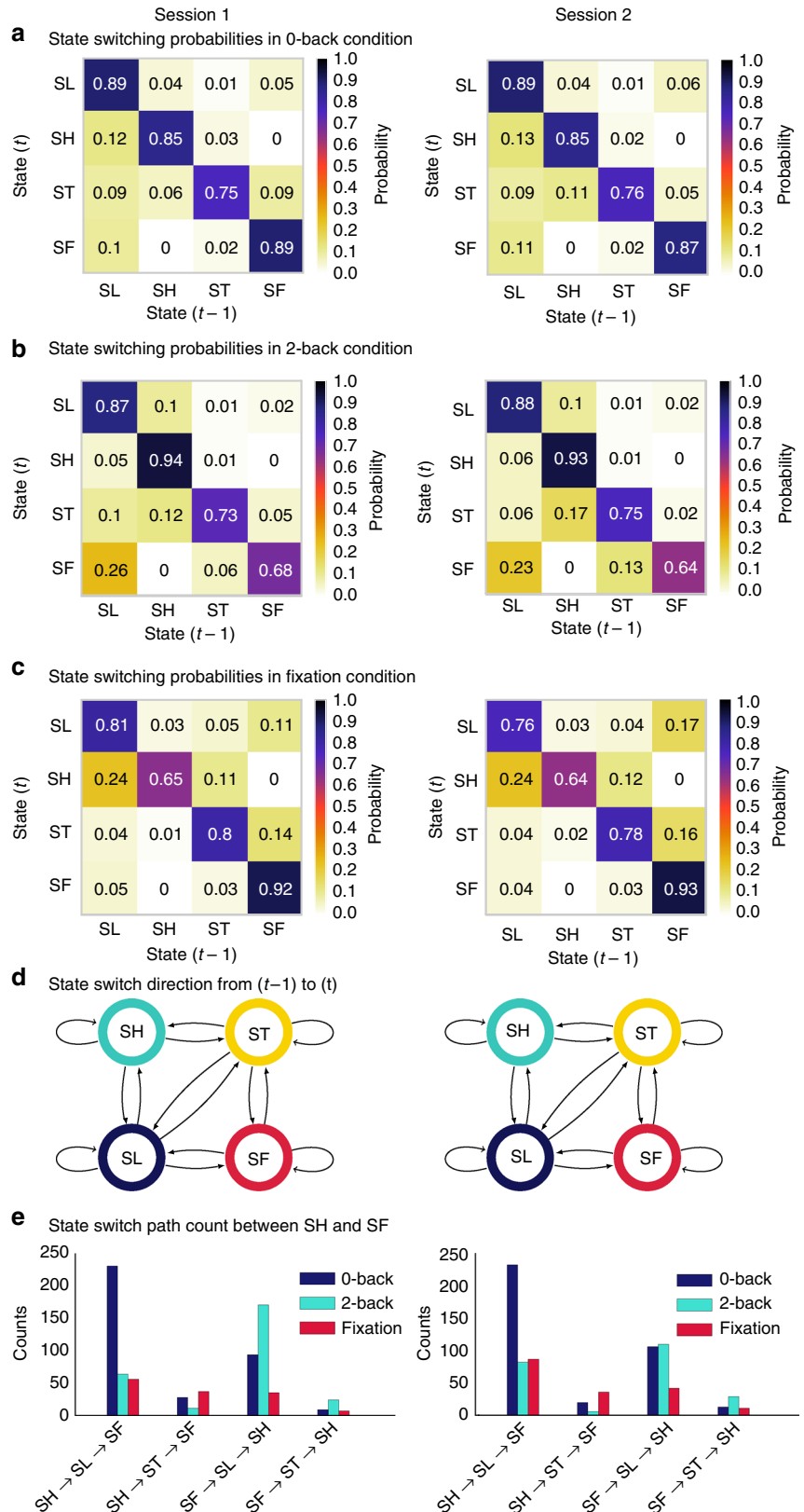

**Fig. 5** Dynamic switching properties of latent brain states. **a–c** State transition probability, defined as the probability that a latent brain state at time instance $t$ stays in its own state or switches to other states at time instance $t - 1$, for each of the four latent states that were identified in the three task conditions. States are persistent over time in each of the task conditions, i.e., they are more likely to stay in their own state rather than to switch to other states in the next time point ($ps < 0.001$, two-tailed $t$-test). **d** State transition probability diagram illustrating that the state SH, which dominates the 2-back task condition, does not switch directly to SF, the state which dominates the fixation condition, and vice versa. As shown in **a–c** the transition probability between these states was zero. **e** Analysis of likely switching paths between SH and SF, revealed that these states first pass through SL and ST

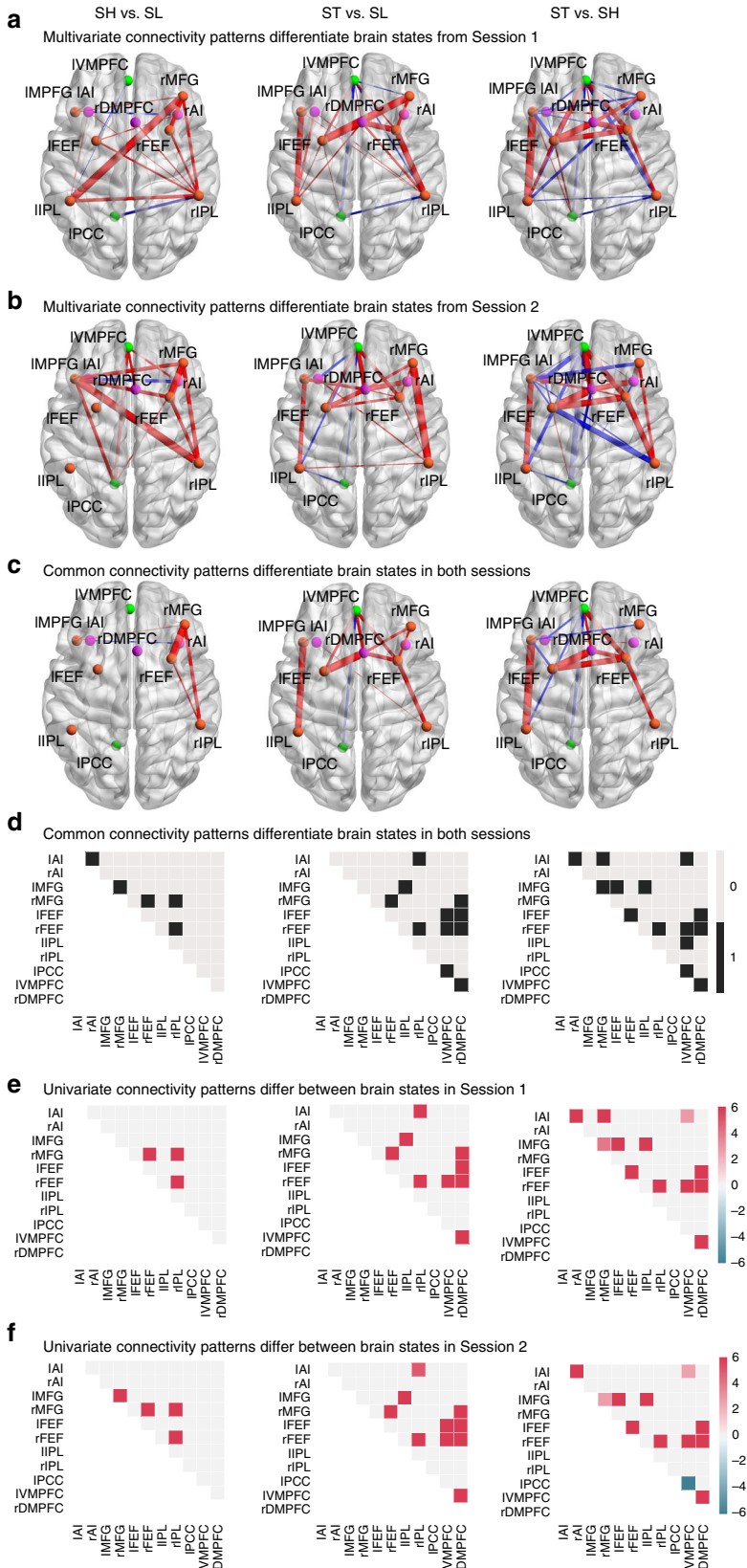

**Fig. 6** Dynamic functional connectivity patterns that distinguish latent brain states. **a**, **b** Brain connectivity patterns that distinguish the latent states SH, SL, and ST in Sessions 1 and 2, respectively. The pink and orange nodes represent SN and CEN ROIs, respectively, activated during WM, while the green nodes represent DMN ROIs deactivated during WM. The red and blue edges represent positive and negative nonzero weights. **c**, **d** Common patterns of connectivity that distinguish latent states in both sessions. **e**, **f** Specific links that showed significant differences in functional connectivity between latent brain states in Sessions 1 and 2, respectively. State SH, which dominates the 2-back WM task, showed higher frontoparietal connectivity compared to the state SL which dominates the 0-back low-load condition. Crucially, it was the transition state ST which had the strongest connectivity. Cells are color-coded and scaled by t-score (ps < 0.05, two-tailed t-test, FDR corrected)

performance in the 2-back block. We found that WM task accuracy was positively correlated with the occupancy rate of the dominant state, SH, in the 2-back WM task condition (all $p$s < 0.001, Pearson's correlation, Fig. 7b). Conversely, the occupancy rate of non-dominant brain states during the 2-back task was associated with poorer performance (Supplementary Table 8). Thus, the dominant state SH is a behaviorally optimal brain state

for 2-back working performance—the more time spent in this brain state the better WM task performance, with more deviations leading to poorer performance.

Next, we investigated the mean lifetime, another key feature of temporal evolution of latent brain states, in relation to WM performance using the same analytic procedures described above. We found that the mean lifetimes of the latent brain states in the

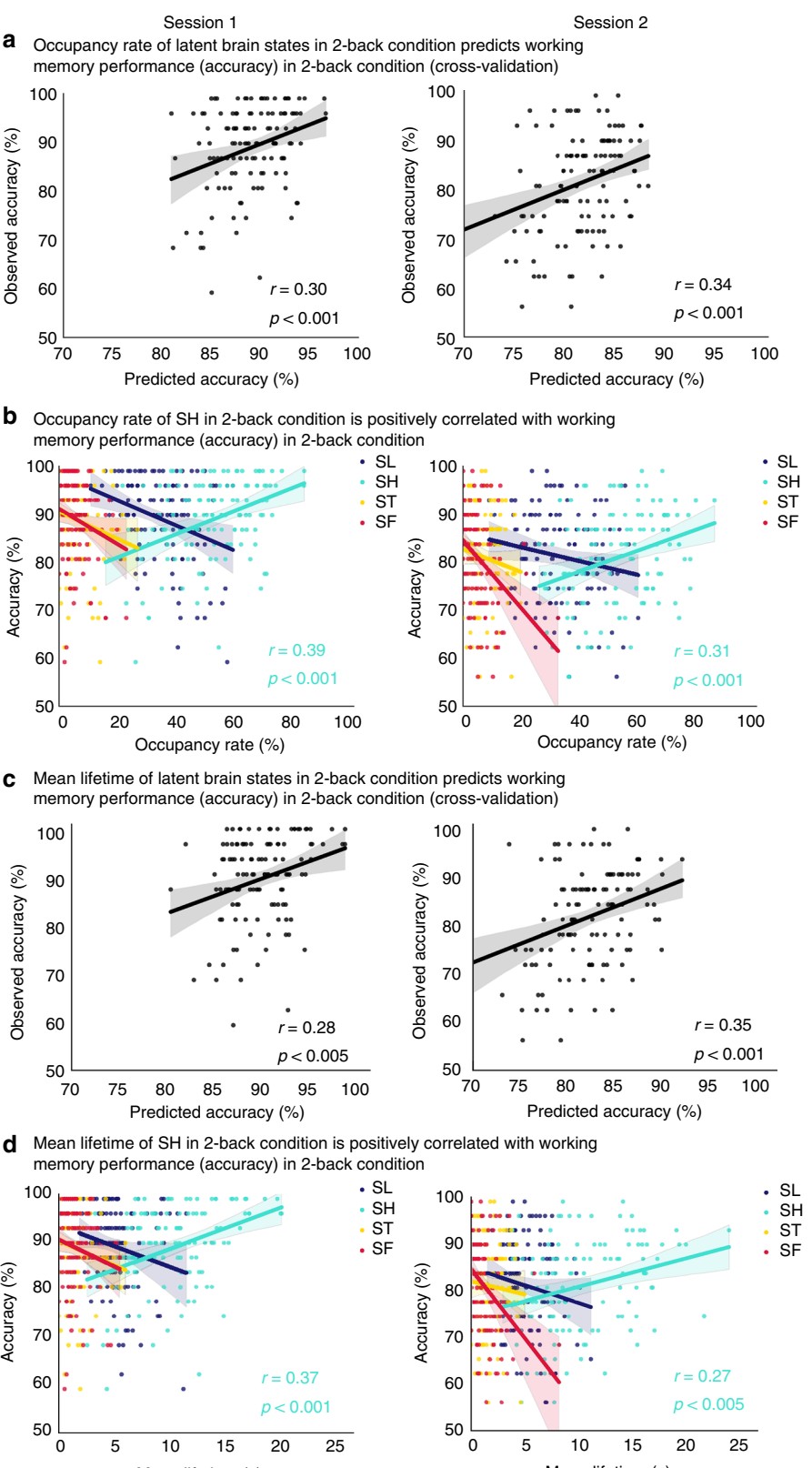

2-back condition predicted WM task accuracy (all $p$s < 0.005, Pearson's correlation, Fig. 7c); furthermore, the mean lifetime of SH in the 2-back task was correlated with 2-back task accuracy (all $p$s < 0.005, Pearson's correlation, Fig. 7d; Supplementary Table 9). Occupancy rate of the state SH was the most robust predictor of WM performance in the 2-back task even after controlling for age, gender and ethnicity (Supplementary Table 10). Finally, we confirmed these results using non-parametric Spearman's correlation (Supplementary Tables 11 and 12). Thus, maintenance of optimal hidden brain states results in better WM task performance.

**Latent brain states predict decision-making dynamics**. To determine whether the optimal brain states, identified above, are associated with efficient decision-making, we used a hierarchical drift-diffusion model to estimate the drift rate, decision threshold and non-decision time based on reaction time data from each participant[41] (Fig. 8a). We found that both the occupancy rate and mean lifetime of the dominant brain state during the 2-back WM task, SH, was correlated with the drift rate, indicating that more the time spent during this state, and the longer the duration of this state, the faster the accumulation of evidence to reach the decision threshold (all $p$s < 0.005, Pearson's correlation, Fig. 8c, e; Supplementary Tables 13 and 14). We confirmed the robustness of this result using a multiple linear regression model with cross-validation in data from both Sessions 1 and 2 (all $p$s < 0.001, Pearson's correlation, Fig. 8b, d). This finding suggests that higher occupancy and longer dwelling time of the dominant brain state contributes to faster and more efficient information processing during the high-load WM condition.

**Robustness of findings with respect to ROI selection**. To examine the robustness of our findings with respect to ROI selection, we conducted a complete set of parallel supplemental analyses using functional clusters from brain networks derived using ICA on resting-state fMRI[42]. All major findings were replicated with this more general resting-state network-derived choice of ROIs (Supplementary Note 1, Supplementary Figs. 5–7, Supplementary Table 15).

**Robustness of findings with respect to head movement**. To test the robustness of our findings with respect to motion, we conducted additional analyses using 12 head motion regression parameters and replicated all the key findings (Supplementary Note 1, Supplementary Figs. 8 and 9, Supplementary Table 16).

**Performance of related HMM-based models WM**. Next, we examined a broad class of HMM-based methods and applied them to opto-fMRI and $n$-back WM fMRI task data, including Hierarchical Dirichlet process hidden Markov model[43], Hierarchical Dirichlet process autoregressive hidden Markov model[15], Hierarchical Dirichlet process switching linear dynamical systems[15] and Bayesian switching factor analysis[29]. These methods were unable to handle model complexity and either overestimated

uncertainties resulting in over-pruning of the latent states thereby converging to a single state, or underestimated uncertainties resulting in multiple states (>10) with little resemblance to the underlying task (Supplementary Note 1; Supplementary Figs. 10–15).

**Performance of temporal clustering techniques during WM WM**. Finally, we examined temporal clustering, an approach that is widely used to investigate dynamic functional connectivity in the human brain[20,44,45]. Briefly, this included: (1) ICA to identify functional clusters from Shier et al.[42] and extracting time series from the ROIs, as described above, (2) applying a sliding window on the time series and estimating time-varying covariance matrices, (3) clustering based on time-varying covariance matrix, and (4) determining the optimal number of clusters. Clustering analysis revealed that the optimal number of clusters was 2 in both data sessions (Supplementary Note 1; Supplementary Fig. 16). Thus, despite the presence of three separate task conditions, only two dynamic brain states could be identified using this approach (Supplementary Note 1; Supplementary Fig. 17). Crucially, we did not find any significant correlation between the occupancy rate of any latent brain states in the 2-back task blocks and WM accuracy ($p > 0.6$, Pearson's correlation, Supplementary Table 17).

## Discussion

Uncovering hidden brain states and their dynamic spatio-temporal evolution in relation to cognitive task demands is a central problem in human brain research. To address this challenge, we developed a state-space approach that identifies moment-by-moment changes in dynamic brain circuits using a rigorous Bayesian Switching Dynamical Systems (BSDS) model. Novel computational tools and unsupervised algorithms allowed us to, for the first time, address critical questions about context-dependent latent brain dynamics, including identification of (i) time-varying functional connectivity, (ii) dynamic functional networks associated with distinct brain states, (iii) the occupancy and mean lifetime of task dominant and sub-dominant brain states, (iv) transition states and their relation to task switching, and (v) brain states that best predict accurate task performance and decision-making dynamics. These analyses allowed us to probe optimal brain states associated with behavior and examine whether the inability to switch between different network configurations impairs cognitive performance. Our findings present a new unsupervised computational model for probing human brain dynamics and time-varying functional interactions that transiently link distributed brain regions during cognition.

BSDS implements an unsupervised learning algorithm which determines latent brain states and dynamic switching processes from observed data. Each brain state is associated with a unique pattern of time-varying functional connectivity in an optimal latent subspace that simultaneously achieves dimensionality and noise reduction. The ensuing brain states are temporally correlated in a Markovian sense—the brain state at a given time

**Fig. 7** Occupancy and mean lifetimes of latent brain states predict WM performance. **a** A multiple linear regression model was trained using occupancy rates of latent brain states in the 2-back task to predict WM accuracy. A significant association was observed and predicted accuracies were correlated with observed accuracy in both sessions: (Session 1: $r = 0.30$, $p < 0.001$; Session 2: $r = 0.34$, $p < 0.001$, Pearson's correlation). **b** Occupancy rate of the latent brain state SH which dominates the 2-back WM task condition was correlated with WM task accuracy in both sessions (Session 1: $r = 0.39$, $p < 0.001$; Session 2: $r = 0.31$, $p < 0.001$, Pearson's correlation). No such relations were found for any of the other latent states. **c** A multiple linear regression model was trained using mean lifetimes of latent brain states in the 2-back task to predict WM accuracy. Here again, a significant association was found and the predicted accuracy was correlated with observed accuracy in both sessions (Session 1: $r = 0.28$, $p < 0.005$; Session 2: $r = 0.35$, $p < 0.001$, Pearson's correlation). **d** Mean lifetime of the latent brain state SH which dominates the 2-back WM task condition was correlated with WM task accuracy in both sessions (Session 1: $r = 0.37$, $p < 0.001$; Session 2: $r = 0.27$, $p < 0.005$, Pearson's correlation). No such relations were found for any of the other latent states. Shaded area represents 95% confidence interval. Color code mapping in **b** and **d**: dark blue—SL, cyan—SH, yellow—ST, red—SF

instance depends on the brain state in the previous time instance. Unlike conventional methods, BSDS does not use arbitrary moving windows or impose temporal boundaries associated with predefined task conditions[4,19,24,46]. Crucially, BSDS learns latent representations and states in a unified framework by optimization of a single objective function within a Bayesian framework using variational inference. The Bayesian framework provides a structured way to automatically regulate model complexity, and the

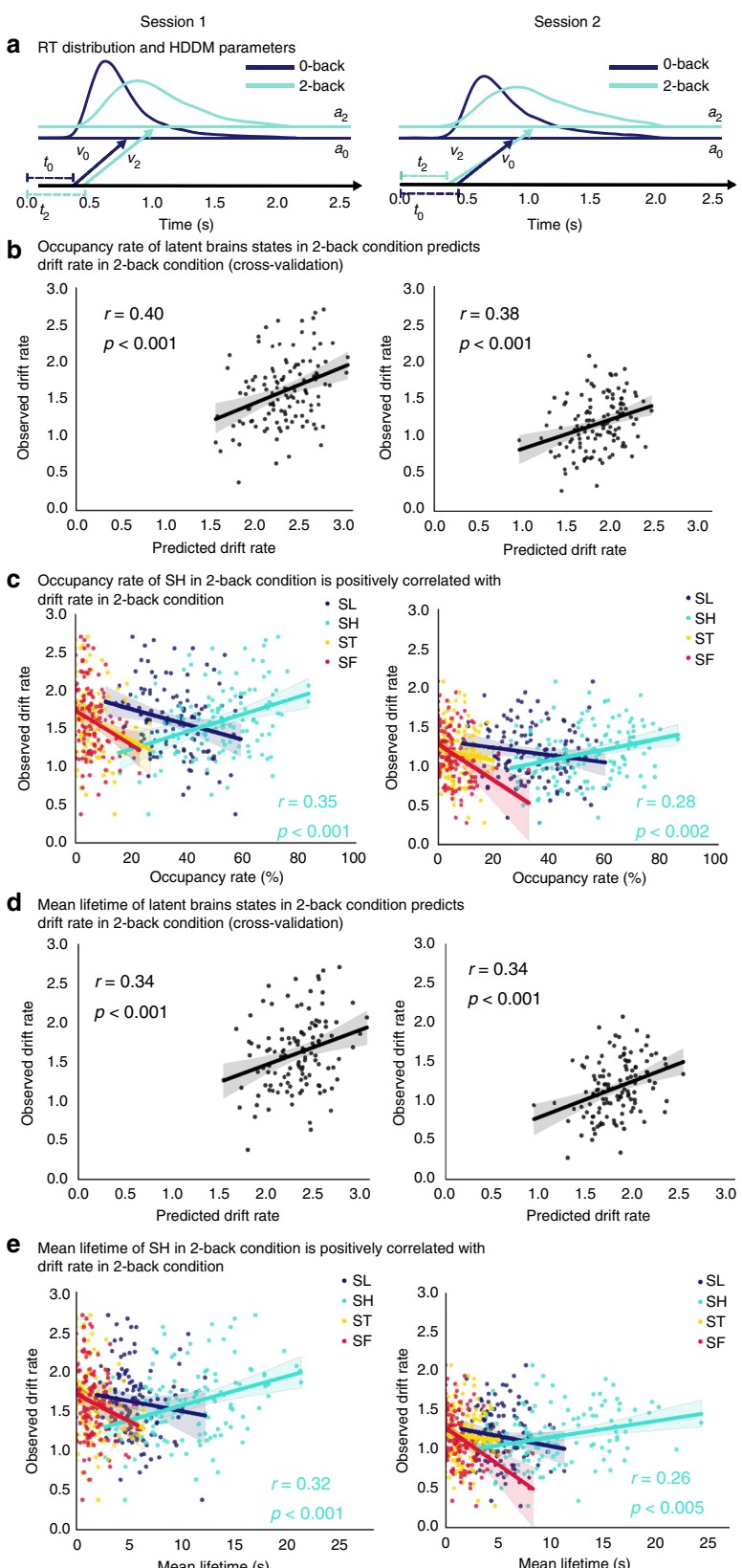

**Fig. 8** Occupancy and mean lifetimes of latent brain state predict decision-making dynamics. **a** Illustration of hierarchical drift-diffusion model (HDDM) and parameter estimates based on the distribution of reaction times (RTs) in the 2-back and 0-back task conditions. **b** A multiple linear regression model was trained using occupancy rates of latent brain states in the 2-back task to predict drift rate in the WM task. A significant association was found, and predicted accuracy was correlated with observed accuracy in both sessions (Session 1: $r = 0.40$, $p < 0.001$; Session 2: $r = 0.38$, $p < 0.001$, Pearson's correlation). **c** Occupancy rate of the latent brain state SH which dominates the 2-back WM task condition was correlated with WM task drift rate in both sessions (Session 1: $r = 0.35$, $p < 0.001$; Session 2: $r = 0.28$, $p < 0.002$, Pearson's correlation). No such relations were found for any of the other latent states. **d** A multiple linear regression model was trained using mean lifetimes of latent brain states in the 2-back task to predict drift rate in the WM task. A significant accuracy was found with predicted accuracy was correlated with observed accuracy in both sessions (Session 1: $r = 0.34$, $p < 0.001$; Session 2: $r = 0.34$, $p < 0.001$, Pearson's correlation). No such relations were found for any of the other latent states. **e** Mean lifetimes of the latent brain state SH which dominates the 2-back WM task was positively correlated with WM task drift rate in both sessions (Session 1: $r = 0.32$, $p < 0.001$; Session 2: $r = 0.26$, $p < 0.005$, Pearson's correlation). No such relations were found for any of the other latent states. Shaded area represents 95% confidence interval. Color code mapping in **c** and **e**: dark blue—SL, cyan—SH, yellow—ST, red—SF

optimal number of latent states and dimensionality of the latent representations are learnt from a formal mathematical model. These features allow BSDS to uncover latent brain states, their temporal evolution, volatility and persistence over time, probability of transition to other brain states, and time-varying functional connectivity in a stable and reliable manner.

To demonstrate that BSDS can accurately learn and identify brain states, we conducted extensive simulations on neurophysiologically-realistic simulations using The Virtual Brain[32] and Neural Engineering Object (Nengo)[47] (see Methods for simulation method details and Fig. 2 for results). A novel test of BSDS, however, comes from application to fMRI data obtained during targeted optogenetic stimulation. In this case, BSDS identified distinct ON and OFF states which overlapped with, but were not defined by, the exact onset and offset of external stimulation. In addition, BSDS uncovered a novel transition state between ON and OFF states. Together, two independent simulation experiments and one opto-fMRI experiment demonstrate the reliable and robust performance of BSDS in uncovering dynamic brain states. Furthermore, BSDS identified dynamic latent states which could not be predicted by stimulation onset and offset alone. Below we show that when applied to human task-related fMRI data this feature not only helps us uncover latent brain states, and their dynamical properties, but also helps identify a novel link between brain states and cognitive task performance.

An important challenge addressed by our study is that brain states during cognition are influenced by unobserved mental processes, such as momentary lapse in attention, changes in motivation, alertness, and fatigue, which can lead to dynamic variations in internal brain states with potentially dramatic negative impact on cognitive performance[12–14]. Crucially, therefore, it cannot be assumed that latent brain states are aligned with task conditions necessitating unsupervised algorithms for estimating dynamic states and the transitions between them. Accordingly, we used a novel computational model and BSDS to probe latent brain states, their fluctuations over time and their relation to experimental manipulation of working memory (WM) load in HCP fMRI data obtained in two different sessions.

Across a large group of 122 participants we detected activation in lateral and medial frontoparietal regions that overlapped with key nodes of the salience, central executive and default mode networks (SN, CEN, and DMN, respectively), three major large-scale neurocognitive networks[48] which underlie a wide range of cognitive functions[31,49–51] including WM[9,10,30,31]. BSDS applied to time series extracted from these regions isolated four latent brain states, each with a distinct covariance pattern associated with frontoparietal, cingulo-opercular, and default mode network regions. Although each task condition was dominated by a distinct brain state (SH, SL, and SF), the four brain states were only partially aligned with onset and offset of the three experimental task conditions (Fig. 4). While the posterior probabilities of individual brain states could accurately predict 2-back, 0-back,

and fixation (rest) task conditions, prediction accuracies were far less than would be expected from a complete alignment with temporal boundaries of task onset and offset. The occupancy rate and mean lifetime of these latent states was much shorter than the duration of the task blocks and each task condition was characterized by a mixture of brain states, with the dominant state active for only a relatively short interval. These results demonstrate that latent (internal) brain states and their associated dynamic functional connectivity patterns are only partially constrained by experimental (external) manipulations. Another novel finding here was the discovery of a fourth transition latent brain state, ST, which was more likely to occur during the transition from one task block to another (Fig. 4). Our identification of such a transition state in the HCP fMRI data mirrors findings from direct optogenetic stimulation noted above. Such novel brain states are difficult to identify without unsupervised probabilistic learning models, such as those implemented in BSDS, which provide a mechanism for capturing latent nonlinear changes in brain connectivity. Together, our results demonstrate that each experimental task condition is characterized by a mixture of dominant, non-dominant and transition states, and that these latent brain states are consistent with, but only weakly defined by, task boundaries. This finding is important because it suggests that previous studies of WM tasks, which have almost always focused on predefined task boundaries to examine functional connectivity, are likely to miss key features of brain state dynamics and the unique functional circuits associated with them. This is most clearly illustrated here by our discovery of the transition state, ST, which would be completely missed by conventional analyses.

Brain states, and their associated functional circuits, are thought to be dynamically reconfigurable to support adaptive cognitive functions. However, the lack of rigorous computational tools has precluded examination of how individual brain states change over time. Specially, we know little about the transition probabilities between likely states and the extent to which latent brain states are persistent over time. BSDS is a generative model that provides a way to address these questions in ways that were virtually impossible heretofore. Examination of moment-to-moment switching between latent brain states in each task condition revealed that while individual brain states do not last over the entire duration of task blocks, these states does not rapidly switch to other states in a random manner (Fig. 5). For all four latent states, transition probabilities were diagonally dominant— they were highest within each state and transitions across states occurred much less frequently. On average, the ratio of switch to non-switch probability was 4:1. Analysis of the state transition probabilities and state switch path further revealed a key constraint on switching between states. Importantly, the dominant brain state in high cognitive load condition (SH) did not directly switch to brain states that dominate the minimal cognitive load condition (SF) or vice versa (Fig. 5). Specifically, transitions

between SH and SF almost always passed through SL the state mostly closely aligned with the intermediate cognitive load condition. These results demonstrate for the first time that the dynamics of brain state switching is constrained by the current brain state as well as changes in cognitive load. Furthermore, not all switch paths are equally likely and there are specific sequences for state switching. As we show below, these dynamic latent brain states features have important behavioral consequences.

Each latent state is associated with a distinct pattern of interregional connectivity, indeed, BSDS relies on changes in network covariance structure to uncover latent brain states. Consistent with this, multivariate analysis of connectivity patterns confirmed high classification rates between each of the four latent brain states. Link-by-link analysis (with FDR correction) identified multiple loci of functional connectivity differences across conditions. The state SH that was dominant during the 2-back high WM load task condition was associated with increased connectivity in the right CEN nodes, specifically frontoparietal connectivity involved the right MFG, FEF, and IPL (Fig. 6). Our findings converge on and extend previous observations based on neuronal activity and brain activation aligned with task onset and offset in WM[9,52,53]. Crucially, while previous studies have shown that structural, intrinsic and task functional connectivity of the dorsolateral frontoparietal circuits play a crucial role in WM performance[54–56], what could not be uncovered previously are changes in functional circuits during the transition state ST, a unique state identified here using BSDS. The transition state ST, showed a unique pattern of extensive differences in cross-network functional connectivity between SN, CEN, and DMN. Specifically, connectivity between the SN and DMN nodes, including AI, DMPFC, and VMPFC, increased significantly during the transition state (Fig. 6). These results emphasize the crucial roles of the SN and DMN in regulating access to attentional resources and modulating large-scale brain networks with changing cognitive task demand[57].

Our analysis not only uncovered unique connectivity patterns associated with the transition state, but also automatically identified that the transition state occurred most frequently right after the onset of a new task block, suggesting that the transition state plays an important, if not critical, role in reconfiguring task-set. Graph-theoretical network analysis of state ST revealed that the DMPFC has the highest node degree and betweenness centrality; furthermore, it was in state ST, that the DMPFC showed the highest node degree and betweenness centrality amongst all four states. Previous research has suggested an important role for this DMPFC region, which overlaps with the pre-supplementary motor areas, in both task control and action selection[58]. However, a recent study using an adaptive control task which required context-sensitive configuration of task-sets and stimulus–response mappings found that activation in the pre-SMA tracked task-set control cost but not response level control[59]. Intracranial recordings have further revealed that neuronal activity in the DMPFC is modulated by task-set but persistent activity in this region was not stimulus-specific[60]. Taken together, these results point to an important role for the DMPFC as a hub for switching internal states in response to changing task conditions.

In sum, our findings provide a dynamic view of brain network reconfiguration during human cognition and highlight novel features of dynamic functional interactions between three core neurocognitive networks during transitions between latent brain states that dominate individual task conditions.

Crucially, the temporal dynamics of latent brain states is not an epiphenomenon; the ability to engage specific brain states has major behavioral implications. Latent factors, such as momentary lapse in attention, have been thought to influence cognitive task performance[12–14] but their effects on brain states and dynamic functional circuits have been hard to characterize. In this context, our study overcomes limitations of previous studies and demonstrates key features of optimal brain states for cognitive task performance.

As noted above, although the 2-back WM task is dominated by a single brain state (SH), this state did not occur for the entire duration of the block during which the task is performed. Rather, participants typically also engaged other non-dominant states in this task block. Furthermore, the onset and duration of latent brain states was highly variable across participants. This variability in engagement of specific states has a strong relation with behavior. Specifically, we found that the occupancy rate as well as the mean lifetime of SH during the 2-back task were positively correlated with accuracy on this task (Fig. 7). In contrast, the occupancy rate and mean lifetime of the other states during the 2-back task was negatively correlated with task accuracy. These results demonstrate that SH is an optimal latent brain state for cognitive performance during the high-load WM task and that deviations from this optimal state impairs performance.

Another novel finding of our study is the link between latent brain states and decision-making. Decision-making during the challenging 2-back WM task involves making trial-by-trial judgments on whether the current stimulus matches a previously seen stimulus, two time points prior. This process is likely susceptible to moment-by-moment variations in perception, attention, and in the ability to dynamically store and refresh the contents of WM. We used a hierarchical drift-diffusion model to determine whether the optimal brain states, identified above, are associated with efficient decision-making. Drift-diffusion models quantify the underlying decision-making process by estimating three latent components underlying the time to respond on each trial: (i) drift rate which indexes how fast evidence is accumulated for a decision, (ii) decision threshold, which indexes the distance to a decision boundary, and (iii) a non-decision time which indexes encoding time prior to decision making[61]. In monkey neurophysiological studies, the drift rate has been linked to cognitive processes during simple two-choice decision tasks[62]. This analysis revealed that higher occupancy and longer dwelling time of the dominant brain state contributes to faster and more efficient information processing during the high-load WM condition. Specifically, engagement and maintenance of the optimal state SH contributes to faster accumulation of evidence allowing individuals to reach the decision threshold whereas shifting to other states results in less efficient decision-making process (Fig. 8).

These results identify brain state properties that are optimal for WM performance and, specifically, identify rapid accumulation of evidence as a key feature associated with this process. Our results also emphasize that maintaining this optimal brain state while performing a high-load cognitive task is beneficial to performance while switching to non-optimal states diminishes performance. Finally, our findings illustrate the power of BSDS and the novel computational model developed here to uncover dynamical features of brain states associated with cognition. Our approach contrasts with previous research, which has assumed a one-to-one correspondence between brain states and WM task conditions, thus missing a crucial link with optimal brain states for behavior. Our analysis thus links the precise temporal boundaries at which states and transitions occur to help disentangle the behavioral significance of temporal fluctuations in dynamic brain connectivity. Whether deviations from the optimal state are related to attentional lapses and changes in arousal[4] is an interesting question for future studies. Dynamic state switching techniques, such as those developed and applied here, provide useful new tools to investigate such questions.

Reproducibility is an important challenge for all of neuroscience research, especially human brain imaging. A recent study reported that statistical power ranges from about 8 to 25% in most neuroimaging studies[63]. The present study makes a unique contribution to the neuroscience literature on human brain dynamics in this regard. Remarkably, as shown in Figs. 4–8, we replicated all key findings in HCP data from Session 2, including identification of dominant and transition brain states, their dynamic spatiotemporal properties, dynamic functional connectivity, and optimal brain states for facilitating cognitive task performance and decision making. Finally, we further demonstrated the robustness of our findings using cross-validation within and across the two sessions. Our study adds to a small but growing body of studies examining reproducibility of dynamic connectivity in the human brain[64].

Earlier approaches for characterizing dynamic interactions have primarily been based on a combination of sliding window and clustering techniques[20,44,45]. These methods rely on ad hoc procedures for determining critical parameters, such as the window length and number of brain states (clusters), which are known to greatly influence the estimation of dynamic brain states and connectivity[19]. The same concern is true of temporal ICA[65] as determining the optimal number of spatial and temporal components is non-trivial. In contrast, BSDS uses a Bayesian framework to automatically regulate model complexity and directly estimates the optimal number of latent states. HMM-based models are well motivated as they are easier to interpret and provide important insights about the states and their temporal evolution, lifetimes, and occurrence of states, which ultimately helps in better characterization of the dynamic brain functional interactions underlying the observed data and their relation to behavioral performance. It should be noted that unlike the present study, previous studies using HMMs for estimation of brain states[27–29] do not explicitly model latent processes underlying observed data. In contrast, BSDS applies HMM to latent space variables generated by an autoregressive process, resulting in greater robustness to abrupt and noisy local changes in the observed data and more robust state identification. Our approach is also an advance over other broad classes of HMM-based methods, including Hierarchical Dirichlet process HMM[43], Hierarchical Dirichlet process autoregressive HMM[15], Hierarchical Dirichlet process switching linear dynamical systems[15], and Bayesian switching factor analysis[29]. We found that these methods underperformed BSDS and failed to accurately identify the underlying brain states. An important challenge for BSDS, and all other HMM-based approaches, is scaling up to a large number of brain regions due to model computational complexity and the limited number of time points in most fMRI studies. Semi-informative initialization of models by bounding the dimensionality of the latent space variables along with interpretable data reduction techniques are needed to address this question.

Our study applied a novel switching dynamical systems approach for investigating fundamental aspects of latent circuit dynamics in the human brain, and their relation to cognition. Leveraging sub-second resolution fMRI data from the HCP we uncovered several novel features of human brain dynamics during WM including dominant and transition brain states, their temporal evolution and dynamic connectivity associated with the SN, CEN, and DMN, three key large-scale brain networks consistently implicated in human cognition. We demonstrate that latent brain states are only weakly aligned with task boundaries and that BSDS not only uncovers task-induced brain states, but also latent transition states and their temporal boundaries during cognition. Thus, a powerful feature of our computational approach is that we can now identify hidden dynamic states that

are important for cognition and which cannot be predicted by stimulus onset and offset alone. This feature allowed us to determine how dynamic latent state transitions allow for flexible reconfiguration of functional circuits in the human brain. Furthermore, we identify latent brain states optimal for task performance and demonstrate that inability to engage and maintain these states in a timely manner is associated with poorer and weaker decision-making dynamics.

More broadly, our study advances computational methods for probing latent dynamical models of human brain function. While there has been an explosion of invasive tools to investigate brain function in animal models, they are of limited use in characterizing functional circuit dynamics in the human brain. As such, our understanding of the behavioral relevance of latent dynamic processes in the human brain has not been adequately addressed. Our report thus fills a critical gap in human neuroscience research. The mathematical framework and computational tools developed here may have wide-ranging applications to the study of aberrant brain dynamics and pervasive cognitive impairments in neuropsychiatric disorders.

## Methods

**Ethics statement**. Data acquisition for the Human Connectome Project was approved by the Institutional Review Board of The Washington University in St. Louis (IRB # 201204036), and all open access data were de-identified.

**Bayesian switching dynamical systems model**. Here, we briefly describe the generative model and inference of the BSDS model. Detailed theoretical derivations are provided in Supplementary Methods. Measures extracted from BSDS include occupancy rate, mean lifetime of latent brain state, temporal evolution of latent brain state, transition probability of latent brain states, state switching probability and mean and covariance of states. Mathematical expression of these measures are discussed in Supplementary Methods.

Let $\boldsymbol{y}_t^s$ denote a $D$-dimensional vector of observed fMRI measurements in time $t$ and for subject $s$. Further, let $\boldsymbol{z}_t^s$ denote a 1-of-$K$ discrete vector of latent state variables of a hidden Markov model (HMM) with elements $z_{kt}^s, \forall k = 1, \ldots, K$. As shown in Supplementary Fig. 1 (top panel), two consecutive time instances are dependent via a first-order Markov chain through an HMM. Specifically, probability distribution of $\boldsymbol{z}_t^s$ depends on the state of the previous latent variable $\boldsymbol{z}_{t-1}^s$ through a conditional distribution $p(\boldsymbol{z}_t^s|\boldsymbol{z}_{t-1}^s, \boldsymbol{A}) = \prod_{k=1}^K \prod_{j=1}^K A_{jk}^{z_{t-1,j}^s z_{tk}^s}$ for all $t>1$ represented by the transition probabilities $\boldsymbol{A}$, where $A_{jk} \equiv p(z_{tk}^s = 1|z_{t-1,j}^s = 1)$, and a marginal distribution $p(\boldsymbol{z}_1^s \mid \boldsymbol{\pi}) = \prod_{k=1}^K \pi_k^{z_{1k}^s}$ represented by a vector of initial probabilities $\boldsymbol{\pi}$ where $\pi_k \equiv p(z_{1k}^s = 1)$[66]. Next, we assume that at a given mode of the system given by the latent state $z_{kt}^s = 1$, observed vector $\boldsymbol{y}_t^s$ is generated via a state-space model in form of:

$$\begin{aligned} \boldsymbol{y}_t^s &= \boldsymbol{U}_k \boldsymbol{x}_{kt}^s + \boldsymbol{\mu}_k + \boldsymbol{e}_{kt}, \quad &\forall t \mid z_{kt}^s = 1, \\ \boldsymbol{x}_{kt}^s &= \bar{\boldsymbol{X}}_{kt}^s \vec{\boldsymbol{V}}_k + \boldsymbol{\epsilon}_{kt}, \quad &\forall t \mid z_{kt}^s = 1. \end{aligned} \tag{1}$$

The first line of the generative model in Eq. (1) can be viewed as a probabilistic factor analysis model[18,67] where $\boldsymbol{U}_k$ is a $D \times P$ dimensional linear transformation matrix that transforms data to a subspace of lower dimensionality, $P < D$, described using a $P$-dimensional vector of latent space variables $\boldsymbol{x}_{kt}^s$ mediated by an overall bias $\boldsymbol{\mu}_k$ and a measurement noise $\boldsymbol{e}_{kt} \sim \mathcal{N}(0, \boldsymbol{\Psi}_k)$. The second line of the generative model can be viewed as an autoregressive (AR) process of order $R$ defined on the latent space variables of the factor analysis model[15]. $\vec{\boldsymbol{V}}_k$ is a vector of AR coefficients. $\bar{\boldsymbol{X}}_{kt}^s = \text{diag}(\bar{\boldsymbol{x}}_{kt}^s)$ is a block diagonal isotropic matrix with elements of $\bar{\boldsymbol{x}}_{kt}^s = (\boldsymbol{x}_{k,t-1}^{s\,\mathrm{T}}, \boldsymbol{x}_{k,t-2}^{s\,\mathrm{T}}, \ldots, \boldsymbol{x}_{k,t-R}^{s\,\mathrm{T}})$ represented using latent space variables from the previous $R$ time frames where T indicates the transpose operator. $\boldsymbol{\epsilon}_{kt} \sim \mathcal{N}(\boldsymbol{m}_k, \boldsymbol{\Sigma}_k)$ models the remaining error term in the latent space. Supplementary Fig. 1 (top panel) shows an AR process of a first-order, $R = 1$ defined on the representations of the observations in the latent subspace, $\boldsymbol{x}_{kt}^s$. Note that all analyses use a first-order autoregressive model, $R = 1$.

A full graphical representation of the generative model is shown in Supplementary Fig. 1. We have introduced some hierarchical parameters, $\nu_{kp}$, which are not explicit in the generative model of Eq. (1). These hyperparameters regulate the model complexity. Detailed description of the model is presented in Supplementary Methods.

We consider a Bayesian treatment of the model. In short, Bayesian inference uses Bayes' theorem to combine priors with data to produce posterior distributions of all model parameters. An exact Bayesian inference for the model described in Eq. (1) and most variants of switching dynamical models is intractable. However,

approximate inference for this family of methods has been developed using Markov chain Monte-Carlo (MCMC) sampling methods in a nonparametric Bayesian framework[15] and using variational inference[68] in a parametric Bayesian framework[16]. Similarly, we consider a parametric Bayesian formulation of the model and use variational inference in order to learn the model parameters and infer the latent variables. Our choice of prior distributions and the inference are discussed in Supplementary Methods.

**Validation experiment—the Virtual Brain model**. To validate BSDS we first used neuronal mass models and investigated whether BSDS can accurately identify brain states and temporal onsets and offsets in simulated data. Simulations were carried out using TVB[32], a state-of-the-art platform for large-scale brain network modeling, as well as custom Python scripts developed by our group. TVB utilizes a dynamic mean field model[33], which can approximate the temporal dynamics of a full network of spiking neurons. Recent work has demonstrated the efficacy of such models in emulating important features of human resting-state activity[34,35]. In this model, the contribution of the inhibitory subpopulation is linearized under the assumption that the typical firing rate of gamma-Aminobutyric (GABA)-ergic interneurons falls in a linear portion of the $f$–$I$ curve, and moreover that the time constant of GABA receptors is significantly shorter than that of NMDA receptors (10 vs 100 ms). Thus, the activity at each node is governed by the following set of stochastic differential equations:

$$
\begin{aligned}
x_i &= w_+ J_E S_i + G J_E \sum_{j=1}^{N} C_{ij} S_j + I_o \\
H(x_i) &= \frac{a x_i - b}{1 - \exp(-d(a x_i - b))} \\
\frac{dS_i}{dt} &= -\frac{S_i}{\tau} + (1 - S_i)\gamma H(x_i) + \sigma v_i(t)
\end{aligned}
\tag{2}
$$

where $x_i$, $H(x_i)$, and $S_i$ denote the driving current, firing rate, and average synaptic gating variable (fraction of open synapses) respectively at each area $i$ within the overall $N$-node network. Anatomical links among brain regions are represented by the structural connectivity matrix $C$ and modulated by the free parameter $G$. Synaptic coupling $J_E$ governs the interactions among regions while $w_+$ further modulates the level of local excitatory recurrence. These parameters are tuned to yield physiologically realistic activity levels across all nodes in a disconnected model. The synaptic gating variable decays with time constant $\tau = 10$ ms. The parameters $a$, $b$, and $d$ of the input-output function $H(x_i)$, as well as the kinetic parameter $\gamma$ are those from[35] where they were optimized to fit numerical solutions of the underlying spiking model. Finally, the Gaussian noise term $\sigma v_i(t)$ is used to drive the system during resting state.

We performed eight simulations of a three-node network using different random initial conditions and noise streams. The following parameters were shared across all simulation runs: raw simulation time step = 0.09765625 ms, sampling rate = 1024 Hz, white matter conduction velocity = 4.0 m/s, noise amplitude $\sigma = 0.0001$ nA, inter-regional connection length = 140 mm. The first 5 s of data from each simulation were discarded to allow for clearance of the initial transients. Local excitatory synaptic activity was converted to BOLD signal via convolution with the canonical hemodynamic response function (HRF) and decimated to TR = 0.7 s using a zero-phase forward–backward filter prior to further analyses. We performed eight simulations of the three-node network using different random initial conditions and noise streams. Each simulation run contained five 60 s ON/OFF task cycles. Task was modeled by stimulating node 1, providing 0.1 nA current with 5 ms pulse width at a rate of 50 Hz for the duration of each 20 s ON block (Fig. 2a).

**Validation experiment—the Neural engineering object model**. Next, we examined whether BSDS could uncover latent states in a more realistic neurobiological circuit with a more complex mix of excitatory–inhibitory interactions. Data for this analysis was generated using neural engineering object (Nengo), a neural simulator that utilizes a large-scale modeling approach[36]. To generate time series data, we used a Nengo-based spiking model of action selection in the cortex-basal ganglia-thalamus circuit with timing predictions that are well matched to both single-cell recordings in rats and psychological paradigms in humans[37]. Ensembles of leaky integrate-and-fire neurons comprised seven nodes of this circuit (basal ganglia: globus pallidus internal, globus pallidus external, subthalamic nucleus, striatum D1, striatum D2; thalamus; motor cortex). To mimic multiple task conditions with individual subject variation, three sufficiently different 16-dimensional input vectors were selected for each individual.

Nengo is a neural simulator that utilizes a large-scale modeling approach[69] based on the Neural Engineering Framework (NEF). By providing a neural compiler, the NEF can take high level algorithms and implement them in neural models that incorporate anatomical and biological restrictions, functional computation, and dynamical systems[70]. A primary principle of the NEF is that ensembles of neurons represent vectors and that connections between these ensembles can compute arbitrary functions on these vectors.

To mimic multiple task conditions with individual subject variation, three sufficiently different 16-dimensional input vectors for each individual were defined

as:

$$
\begin{aligned}
u_1 &\sim \text{Uniform}[0.5, 1.0] \\
u_2 &\sim \text{Uniform}[0.15, 0.45] \\
u_3 &\sim \text{Uniform}[0.1, 0.4] \\
\text{On1} &= [u_1, 1 - u_1, 0, 0, 0, 0, 0, 0, 0, 0, 0, 0, 0, 0, 0, 0] \\
\text{On2} &= [u_2, 1 - u_2, 0, 2, 0, 0, 0, 0, 0, 0, 0, 0, 0, 0, 0, 0] \\
\text{Off} &= [0, 0, 0, 0, 0, 0, 0, 0, 0, 0, 0, 0, u_3, 1 - u_3, u_3, 1 - u_3]
\end{aligned}
$$

Ten simulations (raw simulation time step = 1 ms) were performed for 20 trials of 40 s off and 20 s on (alternating ON1 or ON2) for a total of 1200 s (Fig. 2b). A biological synapse model was applied to filter the output of each of the seven ensembles and then the resulting signal was recorded at each simulation time step. Neural output was converted to BOLD signal by convolving each sample with the canonical HRF at TR = 0.7 s.

**Validation experiment—the optogenetic fMRI data**. Opto-fMRI data, including five adult female Sprague-Dawley rats (250–350 g; Charles River Laboratories, Wilmington, MA), were acquired[38,39]. Two rats were excluded because one did not respond to optical stimulation and the second had movement related artifacts. Of the final three rats included in this study, one was imaged at University of California, Los Angeles (UCLA) and two at Stanford University using identical imaging protocols. During surgery, M1 was targeted and injected with an adeno-associated virus expressing a ChR2-EYFP fusion protein using coordinates −2.7 mm anteroposterior (AP), +3.0 mm mediolateral (ML) right hemisphere, −2.0 mm and −2.5 mm dorsoventral (DV). Additional surgical procedures and details can be found in previous of MRI publications[38].

Experiments were conducted 3 weeks after virus injection for optimal ChR2 expression. The fMRI scans were performed on a 7T small animal MRI system (UCLA: Brucker Biospec, Stanford: Magnex Scientific). All scans used a 39 mm outer diameter and 25 mm inner diameter custom-designed transmit/receive single-loop surface coil. During the fMRI experiment, animals were artificially ventilated under light anesthesia with a mixture of $O_2$ (35%), $N_2O$ (63.5%), isoflurane (1.2–1.5%) and $CO_2$ (3–4%). A block designed fMRI stimulation scheme consisting of six ON–OFF cycles at 20 s ON and 40 s OFF for a total of 6 min was used. During the ON cycles, optical stimulation was delivered at 20 Hz, with a 5 ms pulse duration. The data were acquired using an interleaved spiral readout Gradient Recalled Echo BOLD sequence with 0.5 mm slice thickness and 23 slices. In-plane field of view was designed to be 35×35 mm$^2$ and in-plane spatial resolution was 0.5×0.5 mm$^2$. A sliding window reconstruction was then performed to reconstruct the data into 128×128×23 matrix-size, 750 ms temporal resolution images.

After reconstruction, subject head motion was corrected by the inverse Gauss-Newton motion correction algorithm and 4D fMRI data was analyzed with statistical parameter mapping using the general linear model with five gamma basis. An F-test was then conducted and active voxels were selected as those with corresponding Bonferroni-corrected $p$-values < 0.05. The ROIs were manually selected based on a standard digital rat brain atlas[71] (Fig. 2c).

**Human Connectome Project data**. The Human Connectome Project (HCP) $n$-back WM task fMRI data of 122 individuals (session 1, left-right encoded; session 2, right-left encoded; age: 22–36 years old, 79 female/43 male) were selected from 500 subjects (HCP Q1-Q6 Data Release) based on the following criteria: (1) range of head motion in any translational and rotational direction is less than 1 voxel; (2) average scan-to-scan head motion is less than 0.25 mm; (3) performance accuracy per task block per session is >50%; (4) criterion (1)–(3) must met in both sessions separately; and (5) subjects are right handed.

The HCP $n$-back WM task combines the category specific representation task and the $n$-back WM task in a single task paradigm[40]. Subjects were presented with blocks of trials that consisted of pictures of faces, places, tools, and body parts. Within each session, the four different stimulus type were presented in separate blocks. Furthermore, within each session, half of the blocks are 2-back WM and half are 0-back WM task. In the 2-back WM task blocks, subjects were requested to determine whether the current stimulus matches the stimulus in two presentations of stimuli prior within the same block. In the 0-back WM task blocks, subjects were requested to determine whether the current stimulus matches the target that was presented in the beginning of each block (cue). A 2.5 s cue indicates the task type (and target for 0-back task) at the beginning of each block. Each of the two sessions contains 8 task blocks (10 trials of 2.5 s each, for 25 s) and 4 fixation (rest) blocks (15 s). On each trial, the stimulus is presented for 2 s, followed by a 0.5 s inter-trial-interval (ITI).

**Human fMRI acquisition**. For each individual, 405 frames were acquired in each session using multiband, gradient-echo planar imaging with the following parameters: RT, 720 ms; echo time, 33.1 ms; flip angle, 52°; field of view, 280×180 mm; matrix, 140×90; and voxel dimensions, 2 mm isotropic.

**fMRI preprocessing**. Minimally preprocessed fMRI data for both sessions were obtained from the Human Connectome Project[72]. Spatial smoothing with a Gaussian kernel of 6 mm FWHM was first applied to the minimally preprocessed data to improve signal-to-noise ratio as well as anatomy correspondence between individuals. High-pass temporal filtering ($f > 0.008$ Hz) was applied to remove low frequency signals related to scanner drift.

**General linear model and contrast of interest**. A conventional general linear model (GLM) analysis was conducted in order to determine load-dependent and categorical-dependent activation/deactivation peaks. Each block in each session was modeled as one of the following vector: 0-back-faces, 0-back-places, 0-back-tools, 0-back-body, 2-back-faces, 2-back-places, 2-back-tools, and 2-back-body. The onset and duration of each vector were the onset and duration of the corresponding block. The contrast of interest was WM load effect: 2-back vs 0-back.

**Region of interest and time series**. Load-dependent ROIs were determined on the contrast of interest: 2-back vs 0-back, including 9 load-positive (2-back > 0-back) ROIs: bilateral anterior insula (AI), bilateral middle frontal gyrus (MFG), bilateral frontal eye field (FEF), bilateral intraparietal sulcus (IPS) and dorsomedial prefrontal cortex (DMPFC), and 2 load-negative (2-back < 0-back) ROIs: ventromedial prefrontal cortex (VMPFC) and posterior cingulate cortex (PCC). Each ROI was 6-mm radius sphere centered at the corresponding peak voxel. Time series of the 1st eigenvalue was extracted from each ROI. A multiple linear regression approach with 6 realignment parameters (3 translations and 3 rotations) was applied to time series to reduce head motion-related artifacts and resulting time series was further linearly detrended and normalized.

**Matching BSDS states between sessions**. To determine whether one brain state identified in one session match one brain state identified in another session, we conducted cross-session brain state correlation analysis. Each brain state was defined by a covariate matrix in the latent space, estimated from the set of ROI activation time series in each session separately; and ROI time series can, in turn, be represented as a time series of posterior probabilities of estimated brain states. If a brain state in one session corresponds to a brain state in another session, then time series of posterior probabilities of these two brain states in the same data session should be highly correlated. Specifically, after obtaining four brain states in each session, we first computed posterior probability time series of each brain state in the data session from which brain states are estimated. An example is to compute posterior probability of State $1_1$, estimated from Session 1 data. Next, we computed posterior probability time series of each brain state in the other data session. An example is to compute posterior probability of State $1_2$, estimated from Session 2 data. Then, we computed correlation posterior probability time series of brain states, which were estimated from different sessions, in the same data session. For example, compute correlation between the posterior probability of State $1_1$ in Session 1 and the posterior probability of State $1_2$ in Session 1. High correlation would suggest that State $1_1$ matches State $1_2$ as the two states have highly similar posterior probability in the same data. Indeed, we found exclusive one-to-one mapping between brain states estimated from two data sessions. The Pearson's correlation coefficients between the matched brain states across two sessions range from 0.83 to 0.98 (Supplementary Table 4). To simplify the report and improve the readability, we relabeled the matched brain states in the two sessions to State 1, State 2, State 3, and State 4 in the following data analyses.

**Temporal properties of time-varying latent brain states**. BSDS estimated posterior probability of each latent brain state at each time instance, which allows us to examine several temporal properties of time-varying brain states as the following: (i) the latent brain state with highest posterior probability at a specific time point for a specific subject was chosen as the state at the time point for the subject; (ii) we computed occupancy rate and mean life of each brain state in each task condition (0-back, 2-back, and fixation), which provides task-specific state dominancy information; (iii) we measured state switching probability in each task condition, which quantifies the chance that a brain state at the time instance $t$ stay at its own state or switch to another brain state at the time instance $t + 1$. Some of these temporal properties of latent brain states were then used to examine their relation with task manipulation and behavioral performance as discussed below.

**Time-varying latent brain states predict ongoing task conditions**. To evaluate whether time-varying latent states possesses task manipulation information, we conducted a classification and cross-validation analysis. We used a linear support vector machine from the open-source library LIBSVM (http://www.csie.ntu.edu.tw/~cjlin/libsvm/) to build multiclass classifiers to distinguish task conditions at each time instance. Posterior probability of brain states estimated by BSDS at each time instance were used as features to train classifiers. Classifier performance was evaluated using a within-session and a cross-session cross-validations, separately. First, within-session leave-one-out cross-validation (LOOCV) was conducted in each data session. Specifically, ROI time series and posterior probability time series of the four brain states from one participant was selected as a test set. The rest of the data (training set) were used to train a classifier, which was then applied to the test set to predict which task condition (0-back, 2-back, and rest) should be at each

time instance. This procedure was repeated $S$ times ($S$ is the number of participants) with each participant used exactly once as a test set. The cross-validation accuracy across the test sets was used to evaluate the classifier's performance. The statistical significance of LOOCV accuracy was evaluated using permutation tests (100 times). Second, between-session cross-validation was implemented in the way that the model was trained from one session data and tested on the other session data. Specifically, ROI time series and posterior probability time series of the latent brain states from one session was used as a test set. The other data session was used to train a classifier, which was then applied to the test set to predict task condition at each time instance. The statistical significance of cross-validation accuracy was evaluated using permutation tests (500 times).

**Spatial properties of time-varying latent brain states**. To determine which dynamic functional connections are important for distinguishing different brain states, we conducted feature identification analysis on the covariance matrix derived from BSDS analysis. We first applied logistic regression with Lasso and elastic-net-regularized generalized linear model on feature matrix to distinguish brain states. A 10-fold cross-validation was implemented to optimize lambda for minimizing misclassification error. The optimized lambda was then applied on the full data set and connections with nonzero weights was selected. This multivariate analysis was applied in the two data sessions, separately, and a logical AND operation was used to find common connections with nonzero weights between sessions. Next, we conducted a univariate analysis on the selected common connections and examined which connections are significantly different between brain states in each data session, separately. Last, we further examined whether the common connection patterns are enough to distinguish brain states. To do so, we trained logistic regression classifier only using the common features and LOOCV to evaluate the performance. Static significance of the classifier performance was evaluated using permutation test (500 times).

**Graph analysis of time-varying latent brain states**. To further understand functional network organization in different brain states, we conducted graph analyses using Brain Connectivity Toolbox[73]. First, we converted covariance matrix from BSDS model to Fisher's transformed Pearson's correlation matrix per latent brain state per subject. Then, subject-state-wise z-transformed correlation matrix was binarized so that only edges with top 40% weights were set to 1 while others set to 0[74,75] (results were replicated using other thresholds, e.g., 20, 30, and 50%). Next, we applied Louvain community detection algorithm (gamma = 1) to create subject-state-wise network community[76] and computed node degree and betweenness centrality[73]. Last, we examined interaction effect between node (11 regions) and state (4 latent brain state) and conducted post-hoc $t$-test comparison with Bonferroni correction.

**Time-varying brain state in relation with performance in 2-back task**. To understand the relationship between latent brain states and behavioral performance in the $n$-back task, we first examined whether occupancy rates of latent brain state can predict behavioral performance across participants. Specifically, we built a multiple linear regression model to predict accuracy in the 2-back task based on occupancy rates of brain states in the 2-back task. Performance of prediction model was evaluated using correlation between predicted accuracies and observed accuracies in the 2-back task. Next, used Pearson's correlation to examine the specific relationship between occupancy rate of each brain state in the 2-back task and accuracy in the 2-back task across subjects.

**Time-varying brain state in relation to processing speed in 2-back task**. To further examine whether time-varying brain states impacts information processing and decision making, we used the hierarchical drift-diffusion model (DDM) to extract key parameter of processing speed and conducted brain behavior analysis like above.

The DDM has been extensively used to estimate two-choice decision-making processes[61]. In this framework, decisions are modeled as a combination of three parameters: threshold ($a$) describing the distance between two decision boundaries, drift rate ($v$) describing the rate at which evidence is accumulated for a given decision, and non-decision time ($t$) which is representative of those aspects of response time not included in decision making (e.g., stimulus encoding, movement execution, etc.). Models will sometimes additionally include a decision bias parameter ($z$) if there is a reason a priori to believe that such bias exists in the task data. As there was no such reason for the $n$-back task, we chose not to model decision bias in this study. Here, we estimated the parameters $a$, $v$, and $t$ using the hierarchical DDM (HDDM)[41]. In HDDM, Bayesian inference through Markov chain Monte-Carlo (MCMC) sampling is used to approximate posterior distributions for each parameter at both the individual and group levels. We initialized HDDM to draw 10,000 posterior samples for each of the HCP data sets with the first 1000 samples discarded as burn-in. In order to examine the effect of condition (2-back v 0-back) on $a$, $v$, and $t$, these parameters were estimated separately for 2-back and 0-back trials.

Next, we built a multiple linear regression model to predict drift rate in the 2-back task based on occupancy rates of brain states in the 2-back task. Performance of prediction model was evaluated using correlation between predicted drift rates

and HDDM estimated drift rates in the 2-back task. Then, we performed Pearson's correlation analysis to examine the specific relationship between occupancy rates of each brain state in the 2-back task and drift rates in the 2-back task across subjects.

**Application of BSDS to a relational processing task from the HCP**. To demonstrate that BSDS can reliably estimate dynamic latent brain states in other cognitive tasks, we used fMRI data from a relational processing task that involved matching patterns in visual stimuli. Details of the participant selection, data analysis procedures, and results are in Supplementary Note 1 (Supplementary Figs. 18, 19, Supplementary Table 18).

**Code availability**. Code is available from the authors upon request.

**Data availability**. The *n*-back WM task fMRI data is accessible from the HCP database (https://db.humanconnectome.org/). All of the simulation data are available from the authors.

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

## Acknowledgements

The data used in this study were made publicly available by the Human Connectome Project (http://www.humanconnectome.org/). This research was supported by The Knut and Alice Wallenberg Foundation KAW2014.0392 (J.T.), National Institutes of Health MH105625 (W.C.), HD074652 (S.R.), and EB022907, and NS086085 (V.M.). We thank Drs. Kaustubh Supekar and Aarthi Padmanabhan for useful discussions and valuable feedback on the manuscript.

## Author contributions

J.T., W.C., and V.M. designed the study. J.T., S.R., and V.M. developed the method. J.T., W.C., J.K., J.N. and T.C. performed analysis. J.T., W.C., J.K., and J.N. conducted validation experiments. J.T., W.C. and V.M. wrote and revised the manuscript. J.T., W.C., J. K., J.N., T.C. and V.M reviewed and edited the manuscript.

## Additional information

**Competing interests:** The authors declare no competing interests.

