## [Peer Review File · Nature Communications]

Reviewers' comments:

Reviewer #1 (Remarks to the Author):

Taghia present interesting work evaluating "hidden" states in fMRI data. The paper seems technically rigorous and attempts to shed light on an issue that is coming to the forefront of systems neuroscience – network level time-resolved cognition. The findings match well with work I have been involved in or reviewed (most is not yet published), so I have confidence in their findings though it does raise the question of whether most approaches will in fact converge on the same answer. I was particularly impressed with the finding that the 2-back was characterized by a mixture, but the dominant state was still active for only a short time - which truly supports a non-static view of network activity.

My first overall comment is that this should be two separate papers. A detailed description and validation of the approach (likely going to a more technical journal) and then its application to data to answer a neuroscience relevant question (for Nature Communications or the like). As it stands the paper and its extensive appendix are far too technical and lengthy and does not go into enough depth pressure testing the method or reaching the full capability of the method in explaining ongoing cognition. This would be my recommendation and would allow adding several suggestions below.

Methods portion

-This will need to be reviewed by someone with more experience with Bayesian switching models as I cannot say whether there are any technical deviations on current standard practice.

-How does BSDS deal with noisy data, motion, etc.?

-How do the state results compare to other common time varying approaches?

-How do the timeseries/state weightings/probabilities compare to HMMs or other approaches that allow time dependent weighting (e.g., Vince Calhoun has a PCA approach I believe)

Cognition portion

-This needs to be carried out on the other HCP data. All the same figures don't need to be shown, but high level final summary figures showing that the finding holds across different task types or doesn't – both are interesting and informative. Further, could all task data be combined before running to get a comprehensive state number across tasks?

-A natural question, especially given the cited field in the intro, is what this will detect in rest data. BSDS does not seem to require a fixed task structure (other than for interpretation) and seeing the number of states and visualizing the ones with highest probability would be greatly expand the reach of the paper.

-Is there a relationship to any demographics and state expression during the task?

-Ideally it would have been best to actually have some quantification of "hidden states" such as measures of arousal. Perhaps in a future experiment.

Other comments

Figure 1 needs to be made clearer for a less technically oriented individual. I have some experience in the area and it took me too long to grasp figure 1 and 2 to understand what was going on.

Figure 4a-c: consider changing the color bar. Either to a log scale or have 2 different bars for the diagonal and off diagonal.

-R. Matthew Hutchison

Reviewer #2 (Remarks to the Author):

This paper proposes using a Bayesian switching dynamical systems algorithm to identify latent brain states and characterize their dynamic spatiotemporal properties. The authors apply the proposed method to data from the working memory task of the Human Connectome Projection. In their analysis they identify latent transient brain states and dynamic functional circuits and show that failure to engage these states in a timely manner is associated with poorer task performance. This is very interesting work, and the proposed algorithm looks as if it could potentially be of great use to the field. Below follow some specific comments.

1. The authors should be commended for their use of test-retest data to assess the replicability of the proposed approach. However, also making the code easily available would be beneficial.
2. On page 3, the authors write: "Previous approaches for characterizing dynamic interactions in the human brain have not examined latent processes," I don't think this is technically true as, for example, Vidaurre et al. (2016) used Hidden Markov Models to analyze MEG data. In addition, there are also methods that allow for overlapping states (for example, Leonardi et al. (2014) and Miller et al. (2016)) that seem to imply the presence of latent states. Please comment.
3. On the same page, the authors write: "the application of extant approaches to cognitive task-based fMRI is particularly problematic because they assume that brain states are completely aligned with task onset and offset". It isn't immediately clear to this reviewer why the sliding window and clustering techniques the authors is referring to assume that brain states are aligned with task onset and offset. Please clarify.
4. There are some inconsistencies in the notation used in Figure 1 compared to the rest of the paper. For example, $y(t,s)$ is used instead of y_t^s .
5. It appears from Figure 3a that state S3 occurs in fewer participants than the other states. When it does appear it seems to appear in place of S4. Could it be that they are actually the same state, but split due to between-subject variability?
6. On page 12 the authors discuss reproducibility. It is probably worth mention that several reproducibility studies dealing with dynamic connectivity have recently appeared (see, for example, Choe et al. 2017).
7. In Figure 6 b & d it is difficult to compare the different states as they do not share the same support in the x-axis. Please comment. In addition, could you please clarify how the error bars are computed?
8. The proposed algorithm shares similarities with methods previously used in other fields. For example, the work by Fox et al. cited in the paper. It would be useful to understand what parts of the model are due to the authors, as they claim in the abstract that their method is novel. Perhaps, they mean novel to neuroimaging. In any case this should be clarified.

Reviewer #3 (Remarks to the Author):

The manuscript presents an interesting approach for studying dynamical states in the brain (particularly neural transitions). The approach is first presented, then validated with simulation and optogenetic-fMRI data, before being evaluated with a relatively large task-fMRI dataset focused on working memory performance. The manuscript is generally well written (if a little hard to follow in places) and the methods seem sound. However, I feel that at present it is not clear whether this is a manuscript presenting an important new method or revealing hitherto unknown latent dynamic brain states. It is also not entirely clear how the optogenetic fMRI data relates to the human working memory fMRI data; how the core methodology relates to (and supercedes) other dynamic functional connectivity approaches, and whether the latent brain states identified are task-specific or constitute general neural systems. Therefore, I suggest the authors consider revising the manuscript as follows:

- 1) The presentation of the BSDS algorithm could be much more general way to appeal to the broad audience of Nature Communications. This could involve a simple walk through of how it operates, what is meant by latent state etc and, importantly, a detailed explanation of why and how it is superior to other approaches.
- 2) Relating to 1), the approach (especially with the simulations and optogenetic fMRI) should be compared to other approaches that have been used to study dynamic states in fMRI, including other HMM, temporal ICA or simpler functional connectivity approaches. This would allow the reader to properly evaluate the approach.
- 3) The results from the simulations/optogenetic fMRI could feature more heavily in the results section. Similarly, there could be greater explanation of why those results validate the approach. Just finding results does not show that they are the optimal description of the brain.
- 4) The analyses could be repeated with other tasks (e.g., from the HCP). This should be relatively easy and should make the results more convincing and general.
- 5) More detail could be provided about the stability of results of the human fMRI from different starting assumptions; also were different parcellations/networks examined and if so how do they influence the results; how dependent are the results on preprocessing strategies such as temporal filtering and correction for noise.

Reviewer 1

1.1 “Taghia present interesting work evaluating “hidden” states in fMRI data. The paper seems technically rigorous and attempts to shed light on an issue that is coming to the forefront of systems neuroscience – network level time-resolved cognition. The findings match well with work I have been involved in or reviewed (most is not yet published), so I have confidence in their findings though it does raise the question of whether most approaches will in fact converge on the same answer. I was particularly impressed with the finding that the 2-back was characterized by a mixture, but the dominant state was still active for only a short time - which truly supports a non-static view of network activity.

My first overall comment is that this should be two separate papers. A detailed description and validation of the approach (likely going to a more technical journal) and then its application to data to answer a neuroscience relevant question (for Nature Communications or the like). As it stands the paper and its extensive appendix are far too technical and lengthy and does not go into enough depth pressure testing the method or reaching the full capability of the method in explaining ongoing cognition. This would be my recommendation and would allow adding several suggestions below.”

Response: We are glad that the reviewer found our study to be technically rigorous and one that addresses an important issue at the forefront of human systems neuroscience. We appreciate the reviewer’s suggestion about splitting the manuscript into two papers, and have ourselves grappled with this issue. However, per the editor’s suggestion, we have kept the manuscript as one. To address the reviewer’s concern, and to keep our study focused on latent processes related to working memory, we have relegated most of the methodological and technical details of and additional analyses in the SI, but focused on the scientific questions surrounding latent brain states in the main text. The Supplementary Materials provide details of algorithm, testing, validation and experimental procedures.

1.2 “How does BSDS deal with noisy data, motion, etc.?”

Response: The generative model in BSDS assumes that noise component in the observed data as well as the latent space are Gaussian distributed with a diagonal covariance matrix. Of course, in practice, the exact form of noise is unknown. However, our simulations suggest that on simulated data, where data are generated from the generative model, BSDS can handle the additive noise well (roughly up to about -20dB). We conducted additional analysis (see below) to address this question (**Figure R1**). As with most fMRI studies, motion was regressed from the ROI time series in our analysis. If we see effect of movement on fMRI data as an overall bias which affects all regions in the brain equally, it can be safely captured in the BSDS model via the overall bias parameter (**SI, Section 1.1**). However, in practice, this assumption may not be true. Hence, our recommendation is to regress out the movement as a part of the preprocessing analysis using standard techniques, as was done in our analysis. We emphasize that the movement that affects all ROIs would not influence estimates of model parameters in the BSDS model. To address the reviewer question, we conducted additional analyses (see below).

Noise: We examined the effect of noise on the performance of BSDS in relation to model complexity and estimation of covariance. We created synthetic data using the BSDS generative model with the following parameters: 400 data samples with data dimensionality of 10, three latent states embedded in an intrinsic 5-dimensional space, each latent state associated with a unique mean and covariance specified by a Gaussian distribution, and a random temporal

evolution of states. The noise variance in the data generation was varied with a factor of x0.1, x1, x10, x100, x1000 which corresponds to signal-to-noise ratios: 10 dB to -30 dB with a step size of 10dB.

BSDS was initialized in a noninformative fashion. The initial number of states was set to 10. The intrinsic dimensionality within each state was set to one less than the actual data dimension (i.e., 9). To smooth out the effect of random initializations, the analysis was repeated 10 times.

Figure R1 shows the estimated number of states (top panel) and error in estimation of the covariance matrix averaged across all states and runs (bottom panel).

Note that in most cases, BSDS is able to correctly identify the exact number of states and the estimated covariance matrix agrees well with the true covariance matrix. When the noise variance however dramatically increases, BSDS underestimates the number of states resulting in poor estimation of the covariance matrices.

Movement: We conducted several analyses to address the reviewer’s question. We reran the entire analysis using 12 head motion parameters regression instead of 6 parameters and compared the two results. With 12 head motion parameters regression, we replicated our main findings including (1) there are mixture of latent brain states in each single task block, (2) there are dominant latent brain state in each task condition, (3) BSDS identifies the transition latent brain state that mostly occurs during transition phase, and (4) occupancy rates, mean lifetimes and posterior probability of latent brain states can predict task performance. In summary, we replicated our findings using different motion regression procedures. Details of analysis and results are described in our response to comment #3.6 as well as in the revised Supplementary Material.

Figure R1. Effect of measurement noise on the performance of BSDS. (top panel) The estimated number of states. (bottom panel) Pearson correlation between estimated covariance matrices and true covariance matrices averaged across all states and runs.

Synthetic data were generated using the BSDS generative model. The measurement noise variance was varied with a factor of x0.1, x1, x10, x100, x1000 (~SNRs: 10dB to -30dB). The experiment for each condition was repeated for 10 independent runs with random seed. At each run, data is generated on fly. BSDS was initialized fully noninformative with 10 states.

1.3 “How do the state results compare to other common time varying approaches?”

Response:

We examined the performance of a broad class of HMM based methods and applied them to opto-fMRI and n-back working memory fMRI task data. HMM models can be categorized based on their inference into (i) maximum-likelihood and (ii) Bayesian models. In its standard form, the ML-based HMM model has limited application in practice, as the number of states has to be known in advance. The Bayesian models may be broadly categorized into parametric Bayesian models and nonparametric Bayesian models. Bayesian inference in either forms provides a structured way of handling model complexity, and with sufficient amount of data, they can automatically prune away states with little influence. Another family of methods for modeling temporal data uses autoregressive (AR) process within the HMM framework. In such models, each state of the HMM is assumed to be generated from an AR process. In a more general form, switching linear dynamical models are another important family of models for modeling timeseries. It should be noted that while prior studies have used HMMs for estimation of brain states on the observed data, unlike the present study, they do not explicitly model latent processes underlying observed data. When compared to standard HMM-based models ^{1, 2, 3, 4}, crucially, BSDS applies HMM on the latent space variables, and it is their combination that creates a switching dynamical system which then generates the observed data. This is in direct contrast to previous approaches that have applied HMMs directly to observed MEG ⁴ and resting-state fMRI ^{2, 3}. In comparison, each latent state of BSDS is richer and has greater flexibility due to the autoregressive process on the latent space variables. Conceptually, the system only switches to a new state if the current state is not capable of explaining data. As the result, BSDS shows greater robustness to the abrupt and local changes in data which may not be a part of group pattern of interest. Hence, BSDS often requires fewer states to model the same data when compared to standard HMM based methods. This feature, comparatively, helps in robust identification of states which represent the group effects and ultimately better interpretability of the results.

In the following, we consider multiple other HMM-based methods and the results obtained by applying them to opt-fMRI and n-back working memory data are described below.

Briefly, our analysis of these methods revealed they are unable to properly handle the model complexity by either overestimating uncertainties resulting in over-pruning of the latent states and converging to a single state, or underestimating uncertainties resulting in multiple states (>10) with little resemblance to the task. The latter case arises when models have limited flexibility and are strongly constrained by the Markovian assumptions as the only mechanism for capturing complex dynamical processes hidden in the data. Such models cannot deal with abrupt changes in data which may not be relevant to group patterns of interest. The former case arises in models where each state has extensive degree of flexibility with many parameters to be learnt from data. Although theoretically appealing, in practice, it is difficult to robustly learn all these parameters from noisy data and hence the model behaves too conservatively and can reliably estimate only one state.

In comparison, BSDS takes advantage of the strengths of each while minimizing the weaknesses. Similar to the nonparametric Bayesian switching linear dynamical systems ⁵, it is rich in flexibility. It benefits from both Markovian assumptions and the autoregressive processes in its latent space. However, its flexibility is constrained by using a parametric model with fewer parameters to estimate from the data. On the other hand, it is flexible enough to not react too quickly to every local change in data, allowing it to estimate an optimal and parsimonious set of latent states across study participants.

As requested by the reviewer, we have considered several HMM based methods and applied them on the opto-fMRI and WM datasets. Motivated by our earlier discussion, we considered the following HMM-based methods:

- Category 1: Hierarchical Dirichlet process hidden Markov model (HDP-HMM);
- Category 2: Hierarchical Dirichlet process autoregressive hidden Markov model (HDP-AR-HMM);
- Category 3: Hierarchical Dirichlet process switching linear dynamical systems (HDP-SLDS);
- Category 4: Bayesian switching factor analysis (BSFA).

The above methods cover a broad family of approaches. We then considered various model alternatives in each category. Results and experiment procedure are described in detail below and are included in the Supplementary Results (Section 4).

In summary, on opto-fMRI and WM datasets, both HDP-AR-HMM and HDP-SLDS converged to a single state. BSFA on opto-fMRI resulted in multiple states (14 states) with no clear patterns (**Figure R4**). When BSFA was applied to WM dataset, it converged to 15 and 16 states in Session 1 and 2, respectively (**Figure R7**). HDP-HMM performed well on opto-fMRI (**Figures R2-3**) but resulted in multiple states, more than 10 states, with noisy patterns when applied to WM dataset (**Figure R5-6**).

Performance of HMM-based methods on opto-fMRI and WM datasets

The following summarizes methods and results of HMM-based methods analyses on opt-fMRI and WM datasets.

Methods

We considered the following methods ^{1,3,5}

- Category 1: Hierarchical Dirichlet process hidden Markov model (**HDP-HMM**);
- Category 2: Hierarchical Dirichlet process autoregressive hidden Markov model (**HDP-AR-HMM**);
- Category 3: Hierarchical Dirichlet process switching linear dynamical systems (**HDP-SLDS**);
- Category 4: Bayesian switching factor analysis (**BSFA**).

Categories 1-3 are taken from Emily Fox toolbox, [HDPHMM HDP SLDS toolbox](https://homes.cs.washington.edu/~ebfox/software/) (https://homes.cs.washington.edu/~ebfox/software/). This toolbox implements various HDP-based models ^{1,5}.

Category 4, BSFA ³, can be seen as a special case of BSDS without autoregressive process. [BSFA toolbox](https://github.com/StanfordCosyne/BSFA) (https://github.com/StanfordCosyne/BSFA).

HDP-HMM/HDP-AR-HMM/HDP-SLDS supports time series analysis with various model families. We considered the following model families in our analysis (see [HDPHMM HDP SLDS toolbox](https://homes.cs.washington.edu/~ebfox/software/) for implementation details and refer to ^{1,5} for theoretical discussion):

HDP-HMM

(case1) HDP-HMM.A: Gaussian emissions with non-conjugate Normal-inverse Wishart observation model type shown as N-IW prior.

(case2) HDP-HMM.B: Gaussian emissions with conjugate Normal-inverse Wishart observation model type, shown as NIW prior.

HDP-AR-HMM

(case3) HDP-AR-HMM.A: Autoregressive (AR) order fixed, conjugate matrix-Normal

inverse-Wishart observation model type shown as MNIW prior.

(case4) HDP-AR-HMM.B: Latent AR order with maximal order r with Normal prior together with conjugate matrix-Normal inverse-Wishart Normal observation model type shown as MNIW-N prior.

(case5) HDP-AR-HMM.C: Latent AR order with maximal order r with Normal prior together with non-conjugate Normal inverse-Wishart observation model type shown as N-IW-N prior.

(case6) HDP-AR-HMM.D: Latent AR order with maximal order r with automatic relevance determination observation model type shown as ARD prior.

HDP-SLDS

(case7) HDP-SLDS.A: SLDS with conjugate matrix-Normal inverse-Wishart (MNIW) observation model type, and inverse Wishart (IW) prior on noise.

(case8) HDP-SLDS.B: SLDS with conjugate matrix-Normal inverse-Wishart (MNIW) observation model type, and non-conjugate inverse Wishart Normal (IW-N) prior on noise.

(case9) HDP-SLDS.C: SLDS with conjugate matrix-Normal inverse-Wishart (MNIW) observation model type, and conjugate inverse Normal Wishart Normal (NIW) prior on noise.

(case10) HDP-SLDS.D: SLDS with non-conjugate Normal inverse-Wishart Normal (N-IW-N) observation model type, and inverse Wishart (IW) prior on noise.

(case11) HDP-SLDS.E: SLDS with non-conjugate Normal inverse-Wishart Normal (N-IW-N) observation model type, and non-conjugate inverse Wishart Normal (IW-N) prior on noise.

Method Initializations

Initialization of HDP-HMM/HDP-AR-HMM/HDP-SLDS is fully noninformative as described in HDPHMM HDP-SLDS toolbox. The required number of states is automatically handled within a nonparametric Bayesian learning using MCMC sampling. Initialization of BSFA is fully noninformative based on the default values³. Maximum number of states is set to 20 throughout the analysis. The required number of states is automatically handled within a parametric Bayesian learning using variational inference.

Analysis on Opto-fMRI Dataset

We applied all methods to opto-fMRI dataset to validate their performance on a real dataset for which the ground-truth is known to some extent. From all methods, only three methods could find more than one states, namely HDP-HMM.A,B, BSFA (**Figure R2-R4**). Both HDP-AR-HMM.A-D and HDP-SLDS.A-E converged to only a single state.

HDP-HMM.A-B can be seen as a nonparametric variant of the HMM with Gaussian emission probability distributions. The key difference between HDP-HMM.A and HDP-HMM.B is in their choice of priors. While HDP-HMM.A uses non-conjugate prior model within a fully factorized prior model, HDP-HMM.B uses a conjugate prior model to avoid the full factorization of the priors which theoretically should help with the better recovery of states. Both methods capture temporal correlations in data using Markovian assumptions within an HDP framework. HDP-HMM.A converged to 4 states and HDP-HMM.B converged to 3 states. We also observed in our analysis that while both methods are rather sensitive to the random initialization, HDP-HMM.A appears to be slightly more sensitive in comparison to HDP-HMM.B.

BSFA is a state-space model which uses HMM to capture temporal dependencies as opposed to BSDS which uses both HMM and AR process wrapped into a state-space generative model. As shown in **Figure R4**, the learning converges to several states (14 states) with no clear pattern.

HDP-AR-HMM.A-D use both HMM and AR process in their generative models. We varied the AR order from 1 to 3. All methods consistently converged to a single state. Results are not shown here.

HDP-SLDS.A-E are closest to the BSDS in their generative form but they differ in their inference. While BSDS uses variational inference, they use MCMC sampling. Furthermore, BSDS is a Bayesian parametric model while HDP-SLDS.A-E are Bayesian non-parametric models. In our analysis, we observed all cases converged to a single state. Results are not shown here.

Analysis on HCP Working Memory Dataset

As in the case of opto-fMRI data, HDP-AR-HMM and HDP-SLDS when applied to the WM dataset, converged to a single state. On a side note, we also found HDP-SLDS to be uncomfortably slow for group analysis.

In the following, we direct our attention to HDP-HMM.A and HDP-HMM.B which performed well on the opto-fMRI dataset. We also report the results obtained using BSFA.

HDP-HMM.A,B was initialized as in the case of validation dataset. Estimated temporal evolution of states is shown in **Figures R5, 6**. HDP-HMM.A in Session 1 and 2 converged to 10 and 9 states respectively. Similarly, HDP-HMM.B converged to 10 states in Session 1 and 10 in Session 2.

BSFA was initialized as in the case of validation dataset. Estimated temporal evolution of states is shown in **Figure R7**. BSFA in Session 1 and 2 converged to 15 and 16 states, respectively.

Figure R2. HDP-HMM.A on opto-fMRI dataset. (top) ROI timeseries from the first subject, (middle) stimulation design which is the same for all three subjects, (bottom) estimated temporal evolution of states for each subject. HDP-HMM.A converged to three states.

Figure R3. HDP-HMM.B on opto-fMRI dataset. (top) timeseries, (middle) stimulation design which is the same for all three subjects, (bottom) estimated temporal evolution of states for each subject. HDP-HMM.B converged to three states.

Figure R4. BSFA on opto-fMRI dataset. (top) ROI timeseries from the first subject, (middle) stimulation design which is the same for all three subjects, (bottom) estimated temporal evolution of states for each subject. BSFA was initialized with 20 states and it converged to 14 states.

Figure R5. HDP-HMM.A on WM dataset. (top) ROI timeseries from the first subject, (middle) stimulation design for all subjects, (bottom) estimated temporal evolution of states for each subject. HDP-HMM.A converged to 10 states in Session 1 and to 9 states in Session 2. Note that there is no one-to-one correspondence between color codes across sessions and task designs.

Figure R6. HDP-HMM.B on WM dataset. (top) ROI timeseries from the first subject, (middle) stimulation design for all subjects, (bottom) estimated temporal evolution of states for each subject. HDP-HMM.B converged to 10 states in Session 1 and to 10 states in Session 2. Note that there is no one-to-one correspondence between color codes across sessions and task designs.

Figure R7. BSFA on WM dataset. (top) ROI timeseries from the first subject, (middle) stimulation design for all subjects, (bottom) estimated temporal evolution of states for each subject. BSFA converged to 15 states in Session 1 and to 16 states in Session 2. Note that there is no one-to-one correspondence between color codes across sessions and task designs.

1.4 “How do the timeseries/state weightings/probabilities compare to HMMs or other approaches that allow time dependent weighting (e.g., Vince Calhoun has a PCA approach I believe)”

Response: Please note that the latent component of BSDS uses an HMM model (see also Reply 1.3 for additional clarification). It is not straightforward to directly compare BSDS with temporal ICA and PCA models in our study because these models, e.g., Calhoun’s PCA model, do not provide information about the state transition probabilities, temporal evolution of states, occupancy rate, mean lifetime etc., measures that are crucial in our analysis of latent states underlying working memory. For comparison of BSDS with other HMM models, please refer to our response to the comment 1.3.

1.5 “This needs to be carried out on the other HCP data. All the same figures don’t need to be shown, but high level final summary figures showing that the finding holds across different task types or doesn’t – both are interesting and informative. Further, could all task data be combined before running to get a comprehensive state number across tasks?”

Response: We thank the reviewer for this suggestion. In the main text, we have focused on dynamic brain mechanisms underlying working memory using BSDS. Our methods can be applied to any dataset but the ROIs and hypotheses to be tested will vary from study to study. As such, there is no restriction on whether task or rest data can be studied using BSDS. As requested by the reviewer we have applied BSDS on another HCP dataset: the Relational Processing task and found that BSDS identifies dynamic brain states that are stable and replicable. First, BSDS identified multiple latent brain states in single task blocks and thus demonstrate mixture of latent brain states is a common dynamic feature in cognitive tasks, not limited to n-back working memory task. Second, we found that latent brain states were not equiprobably distributed across different task conditions. For example, state S3 was dominant and had longest mean lifetime in the Match condition while state S4 was dominant and had longest mean lifetime in the Relational condition. Crucially, these states were only partially aligned with onset and offset of the experimental task conditions, and were replicated across sessions. We provide further details of our analysis and results below, and include them in the Supplementary Materials.

Finally, while it is methodologically possible (albeit computationally expensive) to combine different task data and examine brain states across different tasks but interpretation would be problematic as latent brain states are dependent on the choice of regions of interest which vary considerably across cognitive tasks.

Data selection

The Human Connectome Project (HCP) Relational processing task fMRI data of 90 individuals were selected based on the following criteria: (1) range of head motion in any translational and rotational direction is less than 1 voxel; (2) average scan-to-scan head motion is less than 0.25 mm; (3) performance accuracy per session is greater than 50%; (4) criterion (1) – (3) must met in both sessions separately; and (5) subjects are right handed.

Relational Processing task

The Relational Processing task was adapted from a previous study⁶. In this task, there were two task conditions: a relational processing condition and a control matching condition, and stimuli with 6 different shapes and 6 different textures. In the relational processing condition, two pairs of stimuli were presented, with one pair at the top of the screen and the other pair at the

bottom. Participants were told that they should first decide what dimension differs across the top pair of the stimuli (shape or texture) and then decide whether the stimuli at the bottom also differ along the same dimension. In the control matching condition, two stimuli were shown at the top of the screen and one at the bottom with a word in the middle of the screen (either “shape” or “texture”). Participants were told to decide whether the bottom stimulus matched either of the top two stimuli in that dimension. Each participant completed two runs of this task. Each run has 3 relational processing blocks, 3 control matching blocks and 3 16 seconds fixation blocks. Each task block has 5 trials, lasting 18 seconds. Each stimulus was presented for 2800 ms, with a 400 ms ITI.

fMRI acquisition

For each individual, 232 frames were acquired in each session using multiband, gradient-echo planar imaging with the following parameters: RT, 720 ms; echo time, 33.1 ms; flip angle, 52°; field of view, 280 × 180 mm; matrix, 140 × 90; and voxel dimensions, 2 mm isotropic.

fMRI preprocessing

Minimally preprocessed fMRI data for both sessions were obtained from the Human Connectome Project ⁷. Spatial smoothing with a Gaussian kernel of 6mm FWHM was first applied to the minimally preprocessed data to improve signal-to-noise ratio as well as anatomy correspondence between individuals. High-pass temporal filtering ($f > 0.008\text{Hz}$) was applied to remove low frequency signals related to scanner drift.

General linear model and contrast of interest

A conventional general linear model (GLM) analysis was conducted in order to determine relational-processing and matching related activation/deactivation peaks. Each block in each session was modeled as one of the two vectors: relational or match. The onset and duration of each vector were the onset and duration of the corresponding block. The contrast of interest was relational versus match.

Region of interest (ROI) and time series

ROIs were determined on the contrast of interest: relational versus match, including bilateral lateral occipital cortex (LOC), supramarginal gyrus (SMG), angular gyrus (AG), middle frontal gyrus (MFG), frontal pole (FP), medial frontal pole (mFP), right anterior insula (AI) and pre-supplementary motor area (preSMA) (**Figure R8**). Each ROI was 6-mm radius sphere centered at the corresponding peak voxel. Time series of the 1st eigenvalue was extracted from each ROI. A multiple linear regression approach with 6 realignment parameters (3 translations and 3 rotations) was applied to time series to reduce head-motion-related artifacts and resulting time series was further linearly detrended and normalized.

Figure R8. BSDS applied to the Relational Processing task data from the Human Connectome Project. **(a)** Brain regions activated (warm colors) and deactivated (cool colors) during the relational processing, compared to the control matching, condition. **(b)** Regions of interest (ROIs) were determined using activation and deactivation peaks from (a). ROIs activated during relational processing and control matching tasks: 1, left lateral occipital cortex (lLOC); 2, right lateral occipital cortex (rLOC); 3, left supramarginal gyrus (lSMG); 4, right supramarginal gyrus (rSMG); 5, left angular gyrus (lAG); 6, right angular gyrus (rAG); 7, left middle frontal gyrus (lMFG); 8, right middle frontal gyrus (rMFG); 9, left frontal pole (lFP); 10, right frontal pole (rFP); 11, medial frontal pole (mFP); 12, right anterior insula (rAI); and 13, pre-supplementary motor area (preSMA).

Matching BSDS states between sessions

We applied BSDS to probe latent brain dynamics associated with ROIs in two data sessions separately. BSDS isolated five latent brain states in data session 1 and four latent brain states in data session 2. To determine whether one brain state identified in one session match one brain state identified in another session, we conducted cross-session brain state correlation analysis. Each brain state was defined by a covariate matrix in the latent space, estimated from the set of ROI activation timeseries in each session separately; and ROI timeseries can, in turn, be represented as a timeseries of posterior probabilities of estimated brain states. If a brain state in one session corresponds to a brain state in another session, then timeseries of posterior probabilities of these two brain states in the same data session should be highly correlated. Specifically, after obtaining four brain states in each session, we first computed posterior probability timeseries of each brain state in the data session from which brain states are estimated. An example is to compute posterior probability of “State 1₁”, estimated from Session 1 data. Next, we computed posterior probability timeseries of each brain state in the other data session. An example is to compute posterior probability of “State 1₂”, estimated from Session 2 data. Then, we computed correlation posterior probability timeseries of brain states, which were estimated from different sessions, in the same data session. For example, compute correlation between the posterior probability of “State 1₁” in Session 1 and the posterior probability of “State 1₂” in Session 1. High correlation would suggest that “State 1₁” matches “State 1₂” as the two states have highly similar posterior probability in the same data. Indeed, we found exclusive one-to-one mapping between four out of the five latent brain states from the data session 1 and the four latent brain states from the data session 2. The Pearson’s correlation coefficients between the matched brain states across two sessions range from 0.83 to 0.98 (**Table R1**). To simplify the report and improve the readability, we relabeled the matched four brain states in the two sessions to “State 1”, “State 2”, “State 3” and “State 4”, and the fifth brain state in the data session 1 to “State 5”.

Table R1. Correlation of posterior probabilities of latent brain states from training and test sessions.

		Session 2 test state				
		3	8	10	14	
Session 1 training state	2	-0.24	0.65	-0.13	-0.27	
	6	0.06	0.46	-0.1	-0.4	
	8	-0.27	-0.24	0.81	-0.23	
	12	0.74	-0.39	-0.19	-0.17	
	15	-0.37	-0.35	-0.28	0.93	

		Session 1 test state				
		2	6	8	12	15
Session 2 training state	3	-0.26	0.07	-0.26	0.75	-0.38
	8	0.67	0.42	-0.22	-0.4	-0.36
	10	-0.13	-0.09	0.81	-0.17	-0.29
	14	-0.28	-0.39	-0.22	-0.19	0.94

Matched latent state	Session 1	Session 2
1	2	8
2	8	10
3	12	3
4	15	14
	6	

Fractional occupancy of task-dominant latent brain states during task

We compared temporal evolution of latent brain states (**Figure R9a**) and block structure of the experimental task (**Figure R9b**). The latent brain states were only partially aligned with onset and offset of the three experimental task conditions (**Figure R9a, c**). The relational, match and fixation conditions were each dominated by distinct brain states S4, S3 and S1 respectively (**Figure R9d**). The brain state dominant (S4) during the relational task condition had an occupancy rate of $47.5 \pm 15.6\%$ in session 1 and $50.9 \pm 15.6\%$ in session 2. In both Sessions, the occurrence of the dominant state (S4) during the relational task condition was significantly higher than the occurrence of other non-dominant states (all $p < 0.001$). The mean lifetime of the dominant state (S4) in the relational condition was 7 ± 2.6 seconds in session 1 and 7.4 ± 2.7 seconds in session 2, significantly longer than the mean lifetime of the other non-dominant brain states (all $p < 0.001$), but much shorter than the 18 seconds task block (**Figure R9e**). Thus, the relational condition is characterized by a mixture of brain states, with the dominant state active for only a relatively short interval. A similar pattern was observed for the states that were dominant in the match and fixation conditions (**Figure R9d, e**). These results demonstrate that the relational processing task is characterized by latent task-induced states whose fractional occupancy is relatively short compared to the task blocks.

Figure R9. Latent brain states during relational processing task, their dynamic properties and replication across Sessions 1 and 2. **(a)** Temporal evolution of the latent brain states identified in each of the 90 participants. **(b)** Corresponding task waveforms of the three task conditions in the relational processing task – relational, match and fixation blocks – are shown in the same layout. **(c)** Time-varying posterior probability of each latent brain state across participants. **(d)** Occupancy rates of latent brain states for the three states which dominate the relational, match, and fixation task blocks, S4, S3 and S1 respectively. **(e)** Mean lifetimes of latent brain states for the three states which dominate the relational, match and fixation task blocks, S4, S3 and S1 respectively ($p < 0.001$). **(f)** Probability of transition time given a latent brain state.

1.6 “A natural question, especially given the cited field in the intro, is what this will detect in rest data. BSDS does not seem to require a fixed task structure (other than for interpretation) and seeing the number of states and visualizing the ones with highest probability would be greatly expand the reach of the paper.”

Response: The reviewer is right that BSDS does not require a fixed task structure. One could apply BSDS on block task-fMRI (as we have done here with the HCP working memory data), fast event-related task-fMRI, or resting-state fMRI data. However, the focus of our study here was to understand dynamic brain states and circuits underlying human working memory using this novel method. Because interpretation of rest data is more challenging, we have preferred to apply methods to another task (Relational data, please see our response to comment #1.5).

1.7 “Is there a relationship to any demographics and state expression during the task?”

Response: To address the reviewer’s question, we conducted multiple linear regression to predict working memory performance (accuracy) in the 2-back task using posterior probability of the state SH in the 2-back task, age and gender. As shown in **Table R2** below, posterior probability of the state SH remained the most robust predictor of working memory performance in the 2-back task. Gender had significant effect in Session 2 data but the effect was not replicated in Session 1.

Table R2. Multiple linear regression revealed that the posterior probability of the state SH was the most robust predictor of working memory performance in the 2-back task.

	Beta	t value	p value
Session1			
posterior probability of state SH in 2-back	0.24	4.31	< 0.001
Age	0.002	0.95	0.35
Gender	0.03	1.8	0.07
Ethnicity	-0.02	-1.06	0.29
Session 2			
posterior probability of state SH in 2-back	0.27	4.05	< 0.001
Age	-0.001	-0.7	0.48
Gender	0.04	2.3	0.02
Ethnicity	-0.03	-1.25	0.2

1.8 *“Ideally it would have been best to actually have some quantification of “hidden states” such as measures of arousal. Perhaps in a future experiment.”*

Response: We agree. This is a great suggestion, it would be definitely interesting to examine the relationship between hidden states and arousal. Unfortunately, we don't have arousal data for the current data.

1.9 *“Figure 1 needs to be made clearer for a less technically oriented individual. I have some experience in the area and it took me too long to grasp figure 1 and 2 to understand what was going on.”*

Response: We have now provided general description of BSDS methods and algorithm details in the Introduction, which can help understand Figure 1 & 2.

1.10 *“Figure 4a-c: consider changing the color bar. Either to a log scale or have 2 different bars for the diagonal and off diagonal.”*

Response: To make the figure clear, we have now written the actual values into the figure.

Reviewer 2

2.1 *“This paper proposes using a Bayesian switching dynamical systems algorithm to identify latent brain states and characterize their dynamic spatiotemporal properties. The authors apply the proposed method to data from the working memory task of the Human Connectome Projection. In their analysis they identify latent transient brain states and dynamic functional circuits and show that failure to engage these states in a timely manner is associated with poorer task performance. This is very interesting work, and the proposed algorithm looks as if it could potentially be of great use to the field. Below follow some specific comments.”*

Response: We are glad that the reviewer found our work to be interesting. We hope our method will be useful to the field and we will share our code on our lab website and through github.

2.2 *“The authors should be commended for their use of test-retest data to assess the replicability of the proposed approach.”*

Response: We thank the reviewer for the positive feedback.

2.3 *“However, also making the code easily available would be beneficial.”*

Response: We will share our code through our lab website and github.

2.4 *“On page 3, the authors write: “Previous approaches for characterizing dynamic interactions in the human brain have not examined latent processes, ...” I don't think this is technically true as, for example, Vidaurre et al. (2016) used Hidden Markov Models to analyze MEG data. In addition, there are also methods that allow for overlapping states (for example, Leonardi et al.*

(2014) and Miller et al. (2016)) that seem to imply the presence of latent states. Please comment.”

Response: We have now acknowledged the use of HMM models in MEG and EEG studies, as well as resting-state fMRI data. It should be noted that while these studies use HMMs for estimation of brain states on the observed data, unlike the present study, they do not explicitly model latent processes underlying observed data.

2.5. *“On the same page, the authors write: “the application of extant approaches to cognitive task-based fMRI is particularly problematic because they assume that brain states are completely aligned with task onset and offset”. It isn’t immediately clear to this reviewer why the sliding window and clustering techniques the authors is referring to assume that brain states are aligned with task onset and offset. Please clarify.”*

Response: The reviewer is correct that the sliding window can recover states that may or may not be aligned with task onset and offset. We were not referring to sliding window on this issue: we were referring to conventional GLM method which is commonly used to identify brain activity and connectivity pattern associated with specific task condition/blocks. But we agree that this sentence is confusing and we have removed it from the revised manuscript.

2.6 *“There are some inconsistencies in the notation used in Figure 1 compared to the rest of the paper. For example, $y(t,s)$ is used instead of y_t^s .”*

Response: We have now revised the figure.

2.7 *“It appears from Figure 3a that state S3 occurs in fewer participants than the other states. When it does appear it seems to appear in place of S4. Could it be that they are actually the same state, but split due to between–subject variability?”*

Response: We appreciate the reviewer’s observation. To address reviewer’s concern, we took a closer look to temporal evolution of latent brain states in each subject in each session. We confirmed that each latent state occurred in each subject in each session; however, the occupancy rate of the state S3 is less than others. In addition, we have shown that SH (S1) does not directly switch to SF (S4) without passing SL (S2) or ST (S3). If ST (S3) and SF (S4) were the same state, we would not have observed such constraints on switching paths.

2.8 *“On page 12 the authors discuss reproducibility. It is probably worth mention that several reproducibility studies dealing with dynamic connectivity have recently appeared (see, for example, Choe et al. 2017).”*

Response: We thank the reviewer for bringing this study to our attention. We have now cited it in the revised manuscript.

2.9. *“In Figure 6 b & d it is difficult to compare the different states as they do not share the same support in the x-axis. Please comment. In addition, could you please clarify how the error bars are computed?”*

Response: In Figure 6b, the x-axis is occupancy rate of latent brain state (unit is %), whereas, in Figure 6d, the x-axis is mean lifetime of latent brain states (unit is second). So they are different measures in different scales. Error bar indicates 95% confidence interval for the regression estimate, computed using a bootstrap procedure from seaborn.Implot (<https://seaborn.pydata.org/generated/seaborn.Implot.html>). This should help readers to get a better sense of estimation: Where there are few data points the estimated confidence interval is wider reflecting the lack of data.

2.10. *“The proposed algorithm shares similarities with methods previously used in other fields. For example, the work by Fox et al. cited in the paper. It would be useful to understand what parts of the model are due to the authors, as they claim in the abstract that their method is novel. Perhaps, they mean novel to neuroimaging. In any case this should be clarified.”*

Response: We agree with the reviewer that the switching linear dynamical systems (SLDS) models have been around. Although BSDS shares a similar generative model with SLDS, it differs in its inference. While the work of Fox et al. is a nonparametric Bayesian model, BSDS is a parametric Bayesian model. The Bayesian inference in Fox’s SLDS model is based on Markov chain Monte Carlo (MCMC) sampling. While BSDS is based on variational inference which provides a deterministic solution. We have now clarified this in Supplementary Results (Section 4).

We have compared BSDS model with the SLDS model of Fox et al. on opto-fMRI dataset and the WM dataset (see our response to comment #1.3). We considered various model alternatives of SLDS. For details, please refer to our response to the reviewer’s comment 1.3. Overall, we found that the SLDS model results to be not insightful as they converged to a single state both on opto-fMRI and WM datasets (see details in Supplementary Results - Section 4).

Reviewer 3

3.1 *“The manuscript presents an interesting approach for studying dynamical states in the brain (particularly neural transitions). The approach is first presented, then validated with simulation and optogenetic-FMRI data, before being evaluated with a relatively large task-fMRI dataset focused on working memory performance. The manuscript is generally well written (if a little hard to follow in places) and the methods seem sound. However, I feel that at present it is not clear whether this is a manuscript presenting an important new method or revealing hitherto unknown latent dynamic brain states. It is also not entirely clear how the optogenetic FMRI data relates to the human working memory FMRI data; how the core methodology relates to (and supercedes) other dynamic functional connectivity approaches, and whether the latent brain states identified are task-specific or constitute general neural systems. Therefore, I suggest the authors consider revising the manuscript as follows.”*

Response: We are glad that the reviewer found our study to be interesting and sound, and thank the reviewer for the suggestion. Below, we address changes made in response to the reviewer suggestions, noting that the focus of our study was the multisession HCP working memory data and brain states and circuits associated with fronto-parietal cortex regions involved in this task. The optogenetic fMRI and simulations were used to validate and provide insights into how BSDS works, which we hope will be useful to the readers.

3.2 *“The presentation of the BSDS algorithm could be much more general way to appeal to the broad audience of Nature Communications. This could involve a simple walk through of how it operates, what is meant by latent state etc and, importantly, a detailed explanation of why and how it is superior to other approaches”*

Response: We thank the reviewer for this suggestion and have modified the manuscript accordingly.

3.3 *“Relating to 1), the approach (especially with the simulations and optogenetic FMRI) should be compared to other approaches that have been used to study dynamic states in FMRI, including other HMM, temporal ICA or simpler functional connectivity approaches. This would allow the reader to properly evaluate the approach.”*

Response: It is not straightforward to directly compare BSDS with temporal ICA and PCA models in our study because these models, e.g., Calhoun’s PCA model, do not provide information about the state transition probabilities, temporal evolution of states, occupancy rate, mean lifetime, etc, measures that are crucial in our analysis of latent states underlying working memory. For comparison of BSDS with other HMM models, please refer to our response to reviewers’ comments #1.3, 1.4, where we have applied other HMM-based models to opto-fMRI and WM datasets.

3.4 *“The results from the simulations/optogenetic FMRI could feature more heavily in the results section. Similarly, there could be greater explanation of why those results validate the approach. Just finding results does not show that they are the optimal description of the brain.”*

Response: We have now included more details in Results section. The goal of the simulations was to show that our algorithm can recover relevant latent space structure when such structure is known. In the case of optogenetic fMRI, where the ground-truth is not known, it is reasonable to expect that there is a transient state between ON and OFF stimulation. This is indeed what we found and it illustrates a useful feature of BSDS that we were able to investigate in the context of dynamics brains states underlying the transition from low to high working memory load in humans. Ultimately, the validation of the approach may lie in our replication of findings, and the importance influence of latent brain states on behavior.

3.5 *“The analyses could be repeated with other tasks (e.g., from the HCP). This should be relatively easy and should make the results more convincing and general.”*

Response: We thank the reviewer for this suggestion. We have now applied the BSDS to another HCP task fMRI data – the Relational Processing task. Please find details in our response to the comment #1.5 and in the revised Supplementary Material.

3.6 *“More detail could be provided about the stability of results of the human FMRI from different starting assumptions; also were different parcellations/networks examined and if so how do they influence the results; how dependent are the results on preprocessing strategies such as temporal filtering and correction for noise.”*

Response:

Response to different starting assumptions:

First, random initializations were used in BSDS framework. Explicitly, prior distributions were initialized all noninformative. Number of states were set to a large value as BSDS uses automatic model selection to prune away states with little influence. The dimensionality of the latent space was set to one less than the number of ROIs, which corresponds to the fully noninformative initialization of these variables. Because BSDS were estimated on two data sessions separately, they had different initializations. Second, we further tested whether changing maximum number of latent states would have an impact on our main findings. In the original analysis, the maximum number of states was set 10. In the additional data analyses suggested by the reviewer, which was used to address parcellation/network and motion correction issues, the maximum number of states was set 15. As reported below, the main findings were replicated using different starting parameters.

Response to different parcellations/networks:

Our main analysis focused on fronto-parietal ROIs based on peaks of task-related activation and deactivation. To examine the robustness of our findings with respect to ROI selection, we conducted a complete set of parallel supplemental analyses using functional clusters from previously published brain networks derived using ICA on resting-state fMRI⁸. Networks of interest included the salience network (SN), central executive network (CEN), default mode network (DMN) and dorsal attention network (DAN), from which we chose the following fronto-parietal regions: bilateral AI, bilateral DLPFC, bilateral FEF, bilateral PPC, PCC, VMPFC and right DMPFC (**Figure R10**). All other processing steps were identical to the previous analysis. As described in detail below, all major findings were replicated with this more general resting-state network-derived choice of ROIs.

Figure R10. BSDS applied to the n-back working memory task data from the Human Connectome Project. Functional clusters from ICA components (Saliency network, Executive Control network and default mode network) were used as ROIs in the analysis, including bilateral anterior insula (AI), dorsolateral prefrontal cortex (DLPFC), frontal eye field (FEF), posterior parietal cortex (PPC), posterior cingulate cortex (PCC), ventromedial prefrontal cortex (VMPFC) and dorsomedial prefrontal cortex (DMPFC).

Matching BSDS states between sessions

We applied BSDS to probe latent brain dynamics associated with ROIs in two data sessions separately. BSDS isolated five latent brain states in session 1 and session 2. To determine whether one brain state identified in one session match one brain state identified in another session, we conducted cross-session brain state correlation analysis. Using the same state matching algorithm applied in the main analysis (see Material and Methods for details), we found exclusive one-to-one mapping of the five latent brain states between two sessions. The Pearson's correlation coefficients between the matched brain states across two sessions range from 0.56 to 0.87 (**Table R3**).

Table R3. Correlation of posterior probabilities of latent brain states from training and test sessions (the n-back working memory task with functional clusters of ICA components as ROIs).

		Session 2 test state				
		5	9	11	12	14
Session 1 training state	5	0.02	-0.27	-0.26	-0.42	0.82
	9	-0.39	0.87	-0.11	0.08	-0.29
	10	-0.20	0.01	0.58	-0.16	-0.10
	12	0.19	-0.23	0.37	-0.15	-0.15
	14	0.24	-0.28	-0.27	0.56	-0.39
		Session 1 test state				
		5	9	10	12	14
Session 2 training state	5	0.06	-0.39	-0.20	0.15	0.27
	9	-0.29	0.87	0.01	-0.24	-0.28
	11	-0.28	-0.12	0.57	0.39	-0.27
	12	-0.41	0.07	-0.16	-0.11	0.56
	14	0.82	-0.29	-0.11	-0.17	-0.37
		Matched latent state		Session 1	Session 2	
	1	14	12			
	2	5	14			
	3	10	11			
	4	9	9			
	5	12	5			

Fractional occupancy of task-dominant latent brain states during working memory task

The 2-back, 0-back and fixation conditions were each dominated by distinct brain states designated SH (high-load state), SL (low-load state), and SF (fixation state) respectively (**Figure R11**). The brain state SH during the 2-back working memory task had an occupancy rate of $47.1 \pm 6.6\%$ in session 1 and $36 \pm 11.5\%$ in session 2 (**Figure R11d**). In both Sessions, the occurrence of the SH during the 2-back task condition was significantly higher than the occurrence of other non-dominant states (all $ps < 0.001$) except the 5th State in Session 2. Although the 5th State had high occupancy rate in 2-back working memory task in Session 2, it also had high occupancy rate in 0-back working memory task but low occupancy rate in rest. Therefore, the 5th State did not have uniquely significant contribution to the 2-back task. So this state is designated SG (general-control state). The mean lifetime of the state SH in the 2-back condition was 13 ± 5 seconds in session 1 and 9 ± 4 seconds in session 2, significantly longer than the mean lifetime of the other brain states (all $ps < 0.001$), but much shorter than the 27.5 seconds task block (**Figure R11e**). Thus, the 2-back condition is characterized by a mixture of brain states, with the dominant state active for only a relatively short interval. A similar pattern was observed for the states that were dominant in the 0-back and fixation conditions (**Figure R11d, e**). These results demonstrate that the working memory task is characterized by latent task-induced states whose fractional occupancy is relatively short compared to the task blocks.

Identification of a novel transition state

In addition to SH, SL, SF and SG, BSDS uncovered a “transition” state (ST in **Figure R11**). The occupancy rate and mean lifetime of this state during the 2-back, 0-back and fixation conditions was comparably lower than the other states. However, ST was more likely to occur during transition after the onset of new task blocks ($27 \pm 19\%$ in session 1 and $41 \pm 13\%$ in session 2, **Figure R11f**), significantly higher than other latent states in both sessions ($ps < 0.05$) but not to SG in session 1. These results demonstrate that cognitive tasks with multiple conditions are characterized not only by latent task-induced states but also by transition states.

Figure R11. BSDS applied on data of a different set of ROIs revealed latent brain states during working memory, their dynamic properties and replication across Sessions 1 and 2. **(a)** Temporal evolution of the four latent brain states identified in each of the 122 participants. **(b)** Corresponding task waveforms of the three task conditions in the n-back working memory task – 0-back, 2-back and fixation blocks – are shown in the same layout. **(c)** Time-varying posterior probability of each latent brain state across participants. **(d)** Occupancy rates of latent brain states for the three states which dominate the 0-back, 2-back, and fixation task blocks, SL, SH and SF respectively. **(e)** Mean lifetimes of latent brain states for the three states which dominate the 0-back, 2-back and fixation task blocks, SL, SH and SF respectively ($p < 0.001$). **(f)** BSDS revealed a novel transition state ST, which occurs more frequently right after the onset of experimental task blocks than other latent states in both sessions ($p < 0.05$) but not to SG in session 1. Note that SL, SH, ST, SF and SG were named by their task dominance in panel c-f, which correspond to S1, S2, S3, S4 and S5 in panel a.

Latent brain states predict working memory performance

To probe the relation between latent brain states and task performance, we took advantage of a key feature of BSDS, which provides estimates of moment-by-moment changes in brain states and connectivity. We examined whether time-varying brain state changes could predict working memory performance. Specifically, we trained a multiple linear regression model to fit estimated the 2-back accuracy using occupancy rates of brain states in the 2-back condition, applied the model on unseen data to predict accuracy, and evaluated model performance by comparing estimated accuracy and observed value across all the subjects. This analysis revealed a significant relation between predicted and actual accuracy (all $ps < 0.005$, **Figure R12a**). Notably, each of these results was replicated in both Sessions 1 and 2, highlighting the robustness of our brain-behavior findings.

We then tested the hypothesis that the occupancy rate of individual brain states in the 2-back condition is associated with performance in the 2-back block. We found that working memory task accuracy was positively correlated with the occupancy rate of the dominant state, SH, in the 2-back working memory task condition (all $ps < 0.01$, **Figure R12b**). Thus, the dominant state SH is a behaviorally optimal brain state for 2-back working performance – the more time spent in this brain state the better working memory task performance, with more deviations leading to poorer performance.

Next, we investigated the mean lifetime, another key feature of temporal evolution of latent brain states, in relation to working memory performance using the same analytic procedures described above. We found that the mean lifetimes of the latent brain states in the 2-back condition predicted working memory task accuracy (all $ps < 0.05$, **Figure R12c**). Thus, maintenance of optimal hidden brain states results in better working memory task performance.

Figure R12. Occupancy rate and mean lifetimes of latent brain states from BSDS applied on data of a different set of ROIs predict working memory performance: replication across Sessions 1 and 2. **(a)** A multiple linear regression model was trained using occupancy rates of latent brain states in the 2-back task to predict working memory accuracy. A significant association was observed and predicted accuracies were correlated with observed accuracy in both sessions: (Session 1: $r = 0.29$, $p < 0.002$; Session 2: $r = 0.26$, $p < 0.005$). **(b)** Occupancy rate of the latent brain state SH which dominates the 2-back working memory task condition was correlated with working memory task accuracy in both sessions (Session 1: $r = 0.31$, $p < 0.001$; Session 2: $r = 0.25$, $p < 0.005$). No such relations were found for any of the other latent states. **(c)** A multiple linear regression model was trained using mean lifetimes of latent brain states in the 2-back task to predict working memory accuracy. Here again, a significant association was found and the predicted accuracy was correlated with observed accuracy in both sessions (Session 1: $r = 0.22$, $p < 0.05$; Session 2: $r = 0.24$, $p < 0.01$). **(d)** Mean lifetime of the latent brain state SH which dominates the 2-back working memory task condition was correlated with working memory task accuracy in session 2 ($r = 0.20$, $p < 0.05$) but marginally in Session 2 ($p = 0.06$).

In sum, we replicated our main findings using a different head motion regression procedure: 1. BSDS uncovered multiple latent states in single task blocks; 2. BSDS identified task-dominated latent states; 3. BSDS identified transition latent state; and 4. Occupancy rates, posterior probabilities, and mean lifetimes of latent brain states in 2-back working memory task can predict individual performance in 2-back task. We included details of these analysis and results in the Supplementary Material in revision.

Response to temporal filtering:

We applied high-pass temporal filtering ($f > 0.008\text{Hz}$) to remove low frequency signals related to scanner drift, which is a standard practice in analyzing task fMRI data. Temporal filtering with other frequency bands may be used to other data with specific task design, but testing filtering different bandpass frequency is beyond the scope of this study.

Response to correction for noise:

To address the reviewer's question on noise correction, we conducted additional analysis by regressing out 12 motion parameters, including 6 standard parameters (3 translational and 3 rotational movement) and derivatives of these 6 standard parameters, and then repeated the same BSDS analysis using 11 ROIs defined by 2-back vs 0-back contrast. As described in detail below, all major findings were replicated with this choice of movement parameters.

Matching BSDS states between sessions

We applied BSDS to probe latent brain dynamics associated with ROIs in two data sessions separately. BSDS isolated five latent brain states in session 1 and session 2. To determine whether one brain state identified in one session match one brain state identified in another session, we conducted cross-session brain state correlation analysis. Using the same state matching algorithm applied in the main analysis (see Material and Methods for details), we found exclusive one-to-one mapping of the five latent brain states between two sessions. The Pearson's correlation coefficients between the matched brain states across two sessions range from 0.94 to 0.97 (**Table R4**).

Table R4. Correlation of posterior probabilities of latent brain states from training and test sessions (the 12 head motion regression analysis).

		Session 2 test state				
		1	4	7	11	13
Session 1 training state	1	-0.16	-0.18	-0.26	-0.42	0.97
	2	-0.45	-0.11	-0.05	0.94	-0.39
	3	-0.33	-0.09	0.97	-0.20	-0.22
	11	-0.19	0.93	-0.08	-0.18	-0.13
	13	0.94	-0.19	-0.37	-0.27	-0.31

		Session 1 test state				
		1	2	3	11	13
Session 2 training state	1	-0.16	-0.44	-0.33	-0.20	0.94
	4	-0.19	-0.11	-0.09	0.93	-0.19
	7	-0.26	-0.06	0.97	-0.09	-0.37
	11	-0.42	0.94	-0.19	-0.17	-0.25
	13	0.97	-0.38	-0.22	-0.14	-0.31

Matched latent state	Session 1	Session 2
1	2	11
2	1	13
3	11	4
4	3	7
5	13	1

Fractional occupancy of task-dominant latent brain states during working memory task

The 2-back, 0-back and fixation conditions were each dominated by distinct brain states designated SH (high-load state), SL (low-load state), and SF (fixation state) respectively (**Figure R13**). The brain state SH during the 2-back working memory task had an occupancy rate of $37.6 \pm 12.2\%$ in session 1 and $34.7 \pm 11.5\%$ in session 2 (**Figure R13d**). In both Sessions, the occurrence of the SH during the 2-back task condition was significantly higher than the occurrence of other non-dominant states (all $p < 0.001$) except the 5th State. Although the 5th State had high occupancy rate in 2-back working memory task too, it also had high occupancy rate in 0-back working memory task but low occupancy rate in rest. Therefore, the 5th State did not have uniquely significant contribution to the 2-back task. So this state is designated SG (general-control state). The mean lifetime of the state SH in the 2-back condition was 10 ± 4 seconds in session 1 and 8 ± 7 seconds in session 2, significantly longer than the mean lifetime of the other brain states (all $p < 0.001$), but much shorter than the 27.5 seconds task block (**Figure R13e**). Thus, the 2-back condition is characterized by a mixture of brain states, with the dominant state active for only a relatively short interval. A similar pattern was observed for the states that were dominant in the 0-back and fixation conditions (**Figure R13d, R13e**). These results demonstrate that the working memory task is characterized by latent task-induced states whose fractional occupancy is relatively short compared to the task blocks.

Identification of a novel transition state

In addition to SH, SL, SF and SG, BSDS uncovered a “transition” state (ST in **Figure R13**). The occupancy rate and mean lifetime of this state during the 2-back, 0-back and fixation conditions was comparably lower than the other states. However, ST was more likely to occur during transition after the onset of new task blocks ($30 \pm 20\%$ in session 1 and $33 \pm 17\%$ in session 2, **Figure R13**), significantly higher than other latent states in both sessions ($p < 0.05$) except that it is marginally significant compared to SH in session 1 ($p = 0.07$). These results demonstrate that cognitive tasks with multiple conditions are characterized not only by latent task-induced states but also by transition states.

Figure R13. BSDS applied on data using 12 motion parameters regression revealed latent brain states during working memory, their dynamic properties and replication across Sessions 1 and 2. **(a)** Temporal evolution of the four latent brain states identified in each of the 122 participants. **(b)** Corresponding task waveforms of the three task conditions in the n-back working memory task – 0-back, 2-back and fixation blocks – are shown in the same layout. **(c)** Time-varying posterior probability of each latent brain state across participants. **(d)** Occupancy rates of latent brain states for the three states which dominate the 0-back, 2-back, and fixation task blocks, SL, SH and SF respectively. **(e)** Mean lifetimes of latent brain states for the three states which dominate the 0-back, 2-back and fixation task blocks, SL, SH and SF respectively ($p < 0.001$). **(f)** BSDS revealed a novel transition state ST, which occurs more frequently right after the onset of experimental task blocks than other latent states in both sessions ($p < 0.05$) except that it is marginally significant compared to SH in session 1 ($p = 0.07$). Note that SL, SH, ST, SF and SG were named by their task dominance in panel c-f, which correspond to S1, S2, S3, S4 and S5 in panel a.

Latent brain states predict working memory performance

To probe the relation between latent brain states and task performance, we took advantage of a key feature of BSDS, which provides estimates of moment-by-moment changes in brain states and connectivity. We examined whether time-varying brain state changes could predict working memory performance. Specifically, we trained a multiple linear regression model to fit estimated the 2-back accuracy using occupancy rates of brain states in the 2-back condition, applied the model on unseen data to predict accuracy, and evaluated model performance by comparing estimated accuracy and observed value across all the subjects. This analysis revealed a significant relation between predicted and actual accuracy (all $ps < 0.01$, **Figure R14a**). Notably, each of these results was replicated in both Sessions 1 and 2, highlighting the robustness of our brain-behavior findings.

We then tested the hypothesis that the occupancy rate of individual brain states in the 2-back condition is associated with performance in the 2-back block. We found that working memory task accuracy was positively correlated with the occupancy rate of the dominant state, SH, in the 2-back working memory task condition (all $ps < 0.005$, **Figure R14b**). Conversely, the occupancy rate of non-dominant brain states during the 2-back task was associated with poorer performance. Thus, the dominant state SH is a behaviorally optimal brain state for 2-back working performance – the more time spent in this brain state the better working memory task performance, with more deviations leading to poorer performance.

Next, we investigated the mean lifetime, another key feature of temporal evolution of latent brain states, in relation to working memory performance using the same analytic procedures described above. We found that the mean lifetimes of the latent brain states in the 2-back condition predicted working memory task accuracy (all $ps < 0.01$, **Figure R14c**). Thus, maintenance of optimal hidden brain states results in better working memory task performance. In sum, we replicated our main findings using a different head motion regression procedure: 1. BSDS uncovered multiple latent states in single task blocks; 2. BSDS identified task-dominated latent states; 3. BSDS identified transition latent state; and 4. Occupancy rates, posterior probabilities, and mean lifetimes of latent brain states in 2-back working memory task can predict individual performance in 2-back task. We included details of these analysis and results in the Supplementary Material in revision.

Figure R14. Occupancy rate and mean lifetimes of latent brain states from BSDS with 12 motion regression predict working memory performance: replication across Sessions 1 and 2. **(a)** A multiple linear regression model was trained using occupancy rates of latent brain states in the 2-back task to predict working memory accuracy. A significant association was observed and predicted accuracies were correlated with observed accuracy in both sessions: (Session 1: $r = 0.29$, $p < 0.002$; Session 2: $r = 0.31$, $p < 0.001$). **(b)** Occupancy rate of the latent brain state SH which dominates the 2-back working memory task condition was correlated with working memory task accuracy in both sessions (Session 1: $r = 0.33$, $p < 0.001$; Session 2: $r = 0.27$, $p < 0.005$). No such relations were found for any of the other latent states. **(c)** A multiple linear regression model was trained using mean lifetimes of latent brain states in the 2-back task to predict working memory accuracy. Here again, a significant association was found and the predicted accuracy was correlated with observed accuracy in both sessions (Session 1: $r = 0.25$, $p < 0.01$; Session 2: $r = 0.24$, $p < 0.01$). **(d)** Mean lifetime of the latent brain state SH which dominates the 2-back working memory task condition was correlated with working memory task accuracy in session 1 (Session 1: $r = 0.23$, $p = 0.01$) but not in Session 2.

Reference

1. Fox E, Sudderth E, Jordan M, Willsky A. An HDP-HMM for Systems with State Persistence. In: *Proceedings of the 25th international conference on Machine Learning* (ed[^](eds) (2008).
2. Ryali S, *et al.* Temporal Dynamics and Developmental Maturation of Saliency, Default and Central-Executive Network Interactions Revealed by Variational Bayes Hidden Markov Modeling. *PLoS computational biology* **12**, e1005138 (2016).
3. Taghia J, Ryali S, Chen T, Supekar K, Cai W, Menon V. Bayesian switching factor analysis for estimating time-varying functional connectivity in fMRI. *NeuroImage* **155**, 271-290 (2017).
4. Vidaurre D, Quinn AJ, Baker AP, Dupret D, Tejero-Cantero A, Woolrich MW. Spectrally resolved fast transient brain states in electrophysiological data. *NeuroImage* **126**, 81-95 (2016).
5. Fox E, Sudderth E, Jordan M, Willsky A. Nonparametric Bayesian learning of switching dynamical systems. *Advances in Neural Information Processing Systems* **21**, 457-464 (2009).
6. Smith R, Keramatian K, Christoff K. Localizing the rostralateral prefrontal cortex at the individual level. *NeuroImage* **36**, 1387-1396 (2007).
7. Glasser MF, *et al.* The minimal preprocessing pipelines for the Human Connectome Project. *NeuroImage* **80**, 105-124 (2013).
8. Shirer WR, Ryali S, Rykhlevskaia E, Menon V, Greicius MD. Decoding subject-driven cognitive states with whole-brain connectivity patterns. *Cerebral cortex* **22**, 158-165 (2012).

Reviewers' comments:

Reviewer #1 (Remarks to the Author):

The authors have done a remarkable job addressing my concerns and providing additional analysis to bolster their findings. As I indicated in my initial review, the work is very exciting and I hope the method and findings shape how functional brain dynamics are considered. It would have been ideal to see the method used on resting-state data as well, but given the extensive length and figure numbers, perhaps this is best saved for future work. So while still remaining technically challenging (the new description and figures are excellent) I support the manuscript's publication in its current form.

R. Matthew Hutchison

Reviewer #2 (Remarks to the Author):

The authors' did a very thorough job in addressing the comments on the previous version of their manuscript. I have a few outstanding questions that need to be addressed.

1. I am confused by Eq. (2) in the Supplemental Material. It appears there is a Σ_k missing (i.e. the variance of the normal distribution should be $U_k \Sigma_k U_k' + \Phi_k$). This is important as in standard linear dynamical systems this relationship leads to a degeneracy between Σ_k and U_k . For these reasons one typically assumes that (i) Σ_k is diagonal, and (ii) restricts either Σ_k to be identity or the columns of U_k to be unit vectors; see for example Roweis and Ghahramani 1999. It may be the case that the authors' are assuming Σ_k is identity here, in which case Eq. (2) would seem to be correct. However, I don't think this is the case. Please comment.

2. I think I was unclear in how I asked a question last time, so I will try again. In Figure 7B it is difficult to compare the different states as they don't share the same support in the x-axis (i.e. ST only takes small values and SH large values). The same comment holds for Figure 7D. In addition, using regression when the explanatory and response variables are both proportions does not seem appropriate, particularly as many of the values fall around 0.0 and 1.0.

3. The legend for Figure 1 still uses $y(t,s)$ instead of y_t^s

References:

Roweis, S., & Ghahramani, Z. (1999). A unifying review of linear Gaussian models. *Neural computation*, 11(2), 305-345.

Reviewer #3 (Remarks to the Author):

The authors appear to have done a good job in responding to my and the other reviewers' comments in general; however, I couldn't see a marked up version of the revised manuscript, and couldn't assess the changes in the new version (I apologize if I have just missed these).

Assuming the revised ms is as the authors say, I still have a few remaining comments: 1) what circumstances would the proposed technique fail in or is it just better in all circumstances than ICA/PCA/recurrent neural networks/ other HMM approaches and for all data? I don't think I know of any approach that is optimal for every situation; 2) Although ICA/PCA are not directly comparable in all ways, they can be compared to some important aspects of the current analyses e.g., in relationship

to predicting WM performance etc; and stability across individuals etc; 3) Did the authors also consider how well recurrent neural networks (like LSTM) could handle similar data and develop complex descriptions of transitions etc?

Reviewer 1

1.1 “The authors have done a remarkable job addressing my concerns and providing additional analysis to bolster their findings. As I indicated in my initial review, the work is very exciting and I hope the method and findings shape how functional brain dynamics are considered. It would have been ideal to see the method used on resting-state data as well, but given the extensive length and figure numbers, perhaps this is best saved for future work. So while still remaining technically challenging (the new description and figures are excellent) I support the manuscript’s publication in its current form. “

Response: We thank the reviewer for the positive feedback and for recommending our manuscript for publication in NCOMM.

Reviewer 2

2.1 “I am confused by Eq. (2) in the Supplemental Material. It appears there is a Σ_k missing (i.e. the variance of the normal distribution should be $U \Sigma_k U^T + \Phi_k$). This is important as in standard linear dynamical systems this relationship leads to a degeneracy between Σ_k and U_k . For these reasons one typically assumes that (i) Σ_k is diagonal, and (ii) restricts either Σ_k to be identity or the columns of U_k to be unit vectors; see for example Roweis and Ghahramani 1999. It may be the case that the authors’ are assuming Σ_k is identity here, in which case Eq. (2) would seem to be correct. However, I don’t think this is the case. Please comment.”

Response: We agree with the reviewer. First, we would like to confirm that Σ_k has been considered implicitly as identity matrix in Eq.2 to avoid possible identifiability/degeneracy problems, as pointed out correctly by the reviewer. Thus, Eq.2 is correct as it stands. For reasons explained below, we found that it might be less confusing to express Eq.2 without explicitly including Σ_k in the expression, which means effect of Σ_k on the observation level is modeled by an identity matrix.

Role of Σ_k :

Σ_k includes information about the noise variance in the latent space variables. We still need to formally compute its posterior distribution as shown in Eq. 19, in order to robustly estimate parameters involved in the expression of the dynamical process on the latent space variables. However, on the observation level, the effect of Σ_k remains throughout as identity to avoid degeneracy (this explains why Eq.2 was expressed without Σ_k). This approach was shown in our work more effective than, in our opinion, rather strong heuristic approach of forcing Σ_k as identity matrix throughout both the observation and latent space level, which we found results in poor estimation of the sufficient statistics of the latent space variables.

2.2 “I think I was unclear in how I asked a question last time, so I will try again. In Figure 7B it is difficult to compare the different states as they don’t share the same support in the x-axis (i.e. ST only takes small values and SH large values). The same comment holds for Figure 7D. In addition, using regression when the explanatory and response variables are both proportions does not seem appropriate, particularly as many of the values fall around 0.0 and 1.0.”

Response: We are sorry that we misunderstood the original comment. The reason we plotted all the state data in the same figure is to provide readers an overview of the distribution of all the

latent states' occupancy rates and mean lifetimes against task performance during the 2-back task. We could plot separate figures for each state but we thought it would not be as insightful as the overlapped version. We would therefore prefer to keep the current figure format.

To address the reviewer's 2nd point, we applied *Spearman's* correlation on the brain-behavior correlation analyses. We found similar results: occupancy rate of SH during the 2-back task is significantly and positively correlated with accuracy of the 2-back task in both task sessions ($p < 0.001$) and mean lifetime of SH during the 2-back task is significantly and positively correlated with accuracy of the 2-back task in both task sessions ($p < 0.005$), whereas occupancy rates and mean lifetimes of other non-dominant states during the 2-back task are negatively correlated with accuracy of the 2-back task in both task sessions (see **Tables R1 & R2**). These results are now included in the Supplementary Materials section of the revised manuscript.

Table R1. Spearman correlation between occupancy rates of latent brain states in the 2-back task and working memory accuracy.

Latent state	Session 1		Session 2	
	r	p	r	p
SL	-0.27	0.003	-0.13	0.13
SH	0.32	0.001	0.29	0.001
ST	-0.19	0.04	-0.12	0.19
SF	-0.15	0.09	-0.42	0.001

Table R2. Spearman correlation between mean lifetimes of latent brain states in the 2-back task and working memory accuracy.

Latent state	Session 1		Session 2	
	r	p	r	p
SL	-0.2	0.03	-0.09	0.31
SH	0.35	0.001	0.26	0.004
ST	-0.2	0.03	-0.06	0.51
SF	-0.11	0.22	-0.42	0.001

2.3 “The legend for Figure 1 still uses $y(t,s)$ instead of y_t^s ”

Response: We have now corrected this, please see revised Fig.1.

Reviewer 3

3.1 “The authors appear to have done a good job in responding to my and the other reviewers' comments in general; however, I couldn't see a marked up version of the revised manuscript, and couldn't assess the changes in the new version (I apologize if I have just missed these).”

Response: We are sorry that the changes were not highlighted. We have now highlighted all the sections where extensive changes that have been made since the original submission.

3.2 “Assuming the revised ms is as the authors say, I still have a few remaining comments: 1) what circumstances would the proposed technique fail in or is it just better in all circumstances than ICA/PCA/recurrent neural networks/ other HMM approaches and for all data? I don't think I know of any approach that is optimal for every situation”

Response: We agree with the reviewer that no approach is likely to be optimal for every situation. Below we briefly mention two situations which could cause difficulties for the proposed method. It should be noted that, other HMM models would be faced with these same issues

Scalability: handling large number of ROIs

In our analysis, we considered a fully noninformative initialization of the BSDS. Specifically, we considered that the number of latent space variables, shown by P , to be one less than the number of ROIs, shown as D , that is we set $P=D-1$. For time series of length ~ 300 , we found that as long as the number of ROIs is $D < 20$, the model can be safely initialized fully noninformatively, and that the model is able to handle the given complexity as expected. However, in dealing with large number of ROIs, a fully noninformative initialization might be problematic, firstly, due to the computational complexity and secondly due to limited amount of data BSDS may fail to discard all the excess variations in data which may result in difficulties in robust estimation of all parameters and latent variables. In dealing with such scenarios, one might consider semi-informative initialization of the model by bounding the dimensionality of the latent space variables to use a certain percentile of variations in the observations (e.g., 90% of variations in data). In this way, optimal number of latent space variables for each latent state can still be determined automatically from data but they are now bounded.

Handling abrupt changes in fast-transient even-related data

BSDS is a generative model assumes an autoregressive process on the latent space variables that imposes smoothness characteristics which can at times be problematic in handling fast-transient even-related data where there are locally sudden changes which happen infrequently. In such cases, although theoretically the HMM should be able to capture those sudden changes, due to the automatic relevance determination (ARD prior) embedded in our Bayesian inference, the states corresponding to those events may undesirably merge into other states. This is because few samples are associated to those sudden events, and consequently their respective states get regulated heavily due to the small evidence from data. However, we believe that these issues maybe more problematic for EEG/MEG than fMRI, given the inherent smoothness of the latter.

3.3 “Although ICA/PCA are not directly comparable in all ways, they can be compared to some important aspects of the current analyses e.g., in relationship to predicting WM performance etc; and stability across individuals etc”

Response: To address the reviewer’s comment, we conducted additional analysis using ICA in combination with temporal clustering, an approach widely used to investigate dynamic functional connectivity using fMRI^{1, 2, 3}. Briefly, this approach includes: (1) group-wise spatial ICA to identify components of interest, (2) extracting time series of components of interest, (3) applying sliding window on time series and estimating time-varying covariance matrices, (4) clustering based on time-varying covariance matrix, and (5) determining the optimal number of clusters.

Because identifying the optimal number of components in spatial ICA and matching ICA components between two task sessions is non-trivial, instead of running group-wise ICA per task sessions, we used ICA map from an independent study⁴. Recall that the same ICA map was used to test the robustness of our findings and it replicated key results based task defined ROIs (Figure S5 in SI). This minimized the impact of different spatial parcellations when comparing different methods and data across two sessions. The ICA masks included bilateral AI, bilateral DLPFC, bilateral FEF, bilateral PPC, PCC, VMPFC and right DMPFC ROIs. We extracted time series from these regions and estimated dynamic functional interactions using a temporal sliding window approach with an exponentially decaying shape and a window length of 25 seconds (34 TRs), which is shorter than the length of blocks (27.5 seconds), and a sliding step of 0.72 seconds (1 TR). Within each time window, we computed the z-transformed *Pearson’s* correlation between the time-series taken pairwise. This resulted in a time-series of correlation matrices (T x C); here T is the number of time windows and C is number of pairwise interactions among regions at each time point. We then applied a group-wise k-mean clustering to the time-series of correlation matrices, with the number of clusters (k) ranging from 2 to 10. Twenty-five different initializations were used to reduce the chance of getting stuck in local minima. Clustering performance was estimated using the silhouette method and the optimal number of clusters was determined based on maximal silhouette across all the iterations⁵.

Clustering analyses revealed that, based on Silhouette value, the optimal number of clusters was 2 in both data sessions. Thus, despite the presence of three separate task conditions, only two dynamic brain states could be identified using this approach (**Figure R1 & R2**). Further, we did not find any significant correlation between the occupancy rate of any latent brain states in the 2-back task blocks and accuracy of the 2-back task ($p > 0.6$; **Table R3**).

We have included these additional analyses in the Supplementary Materials of the revised manuscript.

Figure R1. The optimal number of temporal clusters was determined using the maximal silhouette obtained across multiple iterations. In both data sessions, the maximal silhouette value was 2. Silhouette is a measure for validating clustering, which evaluates how similar a data point is to its own cluster compared to other clusters. Each color represents a k-mean clustering performance with a random initialization (the number of clusters ranges from 2 to 10).

Figure R2. Two temporal states identified using temporal clustering approach in each data session.

Table R3. Correlation between occupancy rates of latent brain states in the 2-back task and working memory accuracy (temporal clustering approach).

Latent state	Session 1		Session 2	
	r	p	r	p
S1	0.02	0.87	0.04	0.65
S2	-0.02	0.87	-0.04	0.65

3.4 *“Did the authors also consider how well recurrent neural networks (like LSTM) could handle similar data and develop complex descriptions of transitions etc?”*

Response: This is an interesting suggestion. We plan to test such recurrent neural networks in our future studies.

Reference

1. Allen EA, Damaraju E, Plis SM, Erhardt EB, Eichele T, Calhoun VD. Tracking whole-brain connectivity dynamics in the resting state. *Cerebral cortex* **24**, 663-676 (2014).
2. Cai W, Chen T, Szegeletes L, Supekar K, Menon V. Aberrant time-varying cross-network interactions in children with attention-deficit/hyperactivity disorder and the relation to attention deficits. *Biological Psychiatry: Cognitive Neuroscience and Neuroimaging* **In press**, (2017).
3. Rashid B, *et al.* Classification of schizophrenia and bipolar patients using static and dynamic resting-state fMRI brain connectivity. *NeuroImage* **134**, 645-657 (2016).
4. Shirer WR, Ryali S, Rykhlevskaia E, Menon V, Greicius MD. Decoding subject-driven cognitive states with whole-brain connectivity patterns. *Cerebral cortex* **22**, 158-165 (2012).
5. Bellec P, Rosa-Neto P, Lyttelton OC, Benali H, Evans AC. Multi-level bootstrap analysis of stable clusters in resting-state fMRI. *NeuroImage* **51**, 1126-1139 (2010).

REVIEWERS' COMMENTS:

Reviewer #2 (Remarks to the Author):

I am in general satisfied with the authors' response to my previous comments. However, I believe the authors should add the paragraph found below the line "Role of Σ_k :" on page 1 of their response to the supplemental material. This will help guide readers in understanding the model assumptions regarding Σ_k . In the current version I see no discussion of this issue in the manuscript.

Reviewer #3 (Remarks to the Author):

The authors have done an excellent job of responding to my queries and I support publication of the manuscript.

Reviewer 2

I am in general satisfied with the authors' response to my previous comments. However, I believe the authors should add the paragraph found below the line "Role of Σ_k :" on page 1 of their response to the supplemental material. This will help guide readers in understanding the model assumptions regarding Σ_k . In the current version I see no discussion of this issue in the manuscript.

Response: We thank the reviewer for the positive feedback. We have now included our responses regarding "Role of Σ_k :" in the Supplemental Material.

Reviewer #3

The authors have done an excellent job of responding to my queries and I support publication of the manuscript.

Response: We thank the reviewer for recommending our manuscript for publication in NCOMM.